# Regulatory polymorphisms modulate the expression of HLA class II molecules and promote autoimmunity

Prithvi Raj[1†], Ekta Rai[1,2†], Ran Song[1], Shaheen Khan[1], Benjamin E Wakeland[1], Kasthuribai Viswanathan[1], Carlos Arana[1], Chaoying Liang[1], Bo Zhang[1], Igor Dozmorov[1], Ferdicia Carr-Johnson[1], Mitja Mitrovic[3], Graham B Wiley[4], Jennifer A Kelly[4], Bernard R Lauwerys[5], Nancy J Olsen[6], Chris Cotsapas[3], Christine K Garcia[7,8], Carol A Wise[8,9,10,11], John B Harley[12,13], Swapan K Nath[4], Judith A James[4], Chaim O Jacob[14], Betty P Tsao[15], Chandrashekhar Pasare[1], David R Karp[16], Quan Zhen Li[1], Patrick M Gaffney[4], Edward K Wakeland[1*]

[1]Department of Immunology, University of Texas Southwestern Medical Center, Dallas, United States; [2]School of Biotechnology, Shri Mata Vaishno Devi University, Katra, India; [3]Department of Neurology, Yale School of Medicine, New Haven, United States; [4]Arthritis and Clinical Immunology Program, Oklahoma Medical Research Foundation, Oklahoma City, United States; [5]Pole de pathologies rhumatismales, Institut de Recherche Expérimentale et Clinique, Université catholique de Louvain, Bruxelles, Belgium; [6]Division of Rheumatology, Department of Medicine, Penn State Medical School, Hershey, United States; [7]Department of Internal Medicine, University of Texas Southwestern Medical Center, Dallas, United States; [8]Eugene McDermott Center for Human Growth and Development, University of Texas Southwestern Medical Center, Dallas, United States; [9]Department of Orthopaedic Surgery, University of Texas Southwestern Medical Center, Dallas, United States; [10]Sarah M. and Charles E. Seay Center for Musculoskeletal Research, Texas Scottish Rite Hospital for Children, Dallas, United States; [11]Department of Pediatrics, University of Texas Southwestern Medical Center, Dallas, United States; [12]Cincinnati VA Medical Center, Cincinnati, United States; [13]Cincinnati Children's Hospital Medical Center, Cincinnati, United States; [14]Department of Medicine, University of Southern California, Los Angeles, United States; [15]Department of Medicine, University of California, Los Angeles, Los Angeles, United States; [16]Rheumatic Diseases Division, Department of Medicine, University of Texas Southwestern Medical Center, Dallas, United States

*For correspondence: edward.wakeland@utsouthwestern.edu

†These authors contributed equally to this work

Competing interests: The authors declare that no competing interests exist.

**Abstract** Targeted sequencing of sixteen SLE risk loci among 1349 Caucasian cases and controls produced a comprehensive dataset of the variations causing susceptibility to systemic lupus erythematosus (SLE). Two independent disease association signals in the HLA-D region identified two regulatory regions containing 3562 polymorphisms that modified thirty-seven transcription factor binding sites. These extensive functional variations are a new and potent facet of HLA polymorphism. Variations modifying the consensus binding motifs of IRF4 and CTCF in the XL9 regulatory complex modified the transcription of HLA-DRB1, HLA-DQA1 and HLA-DQB1 in a chromosome-specific manner, resulting in a 2.5-fold increase in the surface expression of HLA-DR and DQ molecules on dendritic cells with SLE risk genotypes, which increases to over 4-fold after stimulation. Similar analyses of fifteen other SLE risk loci identified 1206 functional variants tightly

linked with disease-associated SNPs and demonstrated that common disease alleles contain multiple causal variants modulating multiple immune system genes.

## Introduction

Systemic Lupus Erythematosus (SLE) is a complex autoimmune disease resulting from a profound loss of immune tolerance to self-antigens (*Olsen and Karp, 2014*; *Theofilopoulos, 1995a*; *1995b*; *Fairhurst et al., 2006*). The disease initiates with the production of autoantibodies against a spectrum of self-antigens (typically >10 in SLE patients), focused on nucleic acids and nucleic-acid-associated proteins. Disease pathology begins with the deposition of immune complexes in various target tissues, leading to the activation of inflammatory effector mechanisms that damage critical organ systems. Patients with SLE can present with combinations of symptoms, including skin rashes, oral ulcers, glomerulonephritis, neurologic disorders, severe vasculitis, and a distinct form of arthritis (*Tsokos, 2011*). This extensive heterogeneity in clinical presentation presumably reflects variations in the sites of immune complex deposition and induced inflammation among patients, but also suggests that SLE may be a collection of related diseases, rather than a single pathogenic process. A generalized loss in immune tolerance by the humoral immune system and the aberrant activation of inflammatory effector mechanisms at the sites of immune complex deposition, however, are consistent features of SLE.

Susceptibility to SLE is caused by a combination of genetic and environmental factors (*Fairhurst et al., 2006*; *Harley et al., 2009*; *Deng and Tsao, 2010*; *Rai and Wakeland, 2011*). Current thought postulates that a collection of common risk alleles mediates the development of an autoimmune-prone immune system which, when coupled with poorly-defined environmental triggers, becomes dysregulated, leading to the development of autoantibodies and the initiation of disease pathologies. Genome-wide association analyses (GWAS) have identified more than 50 SLE risk loci to date, indicating that susceptibility is quite polygenic (*Harley et al., 2009*; *Nath et al., 2008*; *Harley et al., 2008*; *Kim et al., 2012*; *Graham et al., 2006*; *2008*; *2009*; *Hom et al., 2008*; *Gateva et al., 2009*; *Relle et al., 2015*). A variety of candidate genes have been identified within these risk loci, including: HLA-DR and HLA-DQ class II alleles, IRF5, ITGAM (CD11b), STAT4/STAT1, TNFAIP3, and BLK. The functional effects or 'endophenotypes' that these disease genes contribute to the disease process have not been clearly delineated.

GWAS utilize a dense array of single nucleotide polymorphisms (SNP) to map the positions of risk loci within the human genome to relatively small segments, termed linkage disequilibrium (LD) blocks (typically < 200 Kb in length). Within these LD blocks, recombination is infrequent and polymorphisms form stable combinations or 'haplotypes' that persist within populations for extended periods (*Balding, 2006*; *de Bakker et al., 2005*; *Frazer et al., 2007*). Disease associated 'tagging' SNPs are postulated to be imbedded in specific haplotypes that contain the functional variations that impact disease susceptibility. The characteristics of these functional variations and the endophenotypes that they contribute to disease processes are a poorly described aspect of common disease genetics.

Population sequencing studies have identified extensive variations in both the coding and non-coding regions of the human genome (*Abecasis et al., 2010*; *2012*; *Barreiro and Quintana-Murci, 2010*; *Laval et al., 2010*). The ENCODE consortium has investigated the functional characteristics of non-coding regions in the human genome in detail and have defined a plethora of regulatory elements impacting transcription levels and cell lineage differentiation, including histone associated regions, transcription factor (TF) binding sites, and DNase hypersensitivity clusters (*Gerstein et al., 2012*). A parallel series of investigations by several research groups have used expression quantitative trait locus (eQTL) analysis to identify common polymorphisms that quantitatively impact gene transcription (*Sheffield et al., 2013*; *Vernot et al., 2012*; *Dunham et al., 2012*; *Bernstein et al., 2010*; *Cookson et al., 2009*; *Fairfax et al., 2012*; *2014*; *Gilad et al., 2008*). These findings, coupled with data indicating that many disease-tagging SNPs are localized to non-coding regulatory regions (*Maurano et al., 2012*), suggest that the causal variants for common disease risk alleles may impact regulatory processes, rather than protein structure.

**eLife digest** The human immune system defends the body against microbes and other threats. However, if this process goes wrong the immune system can attack the body's own healthy cells, which can lead to serious autoimmune diseases.

Systemic lupus erythematosus (SLE) is an autoimmune disease in which immune cells often attack internal organs – including the kidneys, nervous system and heart. Over the past decade, multiple genes have been linked with an increased risk of SLE. However, it is largely unknown how the sequences of these genes differ between individuals with SLE and healthy individuals, and the precise changes that lead to an increased risk of SLE are also not clear.

Now, Raj, Rai et al. have determined the genetic sequences of over 700 people with SLE and over 500 healthy individuals and looked for differences that influence susceptibility to the disease. The vast majority of differences were discovered in stretches of DNA that regulate the expression of nearby genes, rather than in DNA that encodes the structures of proteins. Notably, extensive differences were found in a region of the human genome that regulates the production of proteins called Human Leukocyte Antigen class II molecules; which are known to play a critical role in activating the immune system. Raj, Rai et al. found that slight changes to the regulatory DNA sequences resulted in an overabundance of these proteins, which led to a hyperactive immune system that is strongly associated with SLE.

Future studies could now ask if the changes to the regulatory DNA sequences highlighted by Raj, Rai et al. increase susceptibility to other autoimmune disorders as well. It may also be possible to use the increased understanding of how the immune system is regulated to develop new ways to minimize the rejection of organ transplants.

Here we describe the targeted, deep sequencing of 28 risk loci for SLE in a population of SLE patients and controls. Our sequencing study identified 124,552 high quality sequence variants contained in these risk loci among 1349 Caucasian cases (773) and controls (576). Detailed analysis of sixteen of these SLE risk loci demonstrate that haplotypes of functional variations in tight LD with SLE- tagging SNPs often impact the expression of multiple genes, resulting in the association of several transcriptional variations with SLE risk haplotypes. Notably, multiple SLE risk haplotypes within the HLA-D region were found to coordinately upregulate HLA-DR, -DQ and a variety of other genes within the antigen processing and presentation pathways for HLA class I and class II molecules. These results reveal a new functional diversification mediated by HLA-D polymorphisms and provide important insights into the molecular mechanisms by which HLA-D and other SLE risk loci potentiate disease.

## Results

### Deep sequencing of SLE risk loci in populations of SLE cases and controls

Targeted genomic sequencing of twenty-eight GWAS-confirmed SLE risk loci was performed using Illumina (Illumina Inc., San Diego, CA) custom enrichment arrays on genomic DNA from 1775 SLE patients and controls (*Supplementary file 1A*, *Figure 1A–F*). These procedures resulted in >128 fold coverage of the genomic segments containing these SLE risk loci (*Supplementary file 1B*, *Figure 1G*). Our bioinformatics pipeline (*Figure 1H*) defined 1349 samples of European American (EA) ancestry (*Figure 2A*) that carried 124,552 high quality variants of which 114,487 are single nucleotide variations (SNVs) and 10,065 are insertion/deletions (In/Del). This sequence-based variant database, which identifies an average of one variant every 39 basepairs in the targeted regions, provides a comprehensive assessment of genomic diversity at SLE risk loci in the EA population. The functional properties of these variants were annotated using multiple databases cataloguing the functional properties of human genomic variation (*Figure 2B,C*), including the phase 3 release from the 1000 genome study (*Auton et al., 2015*), the PolyPhen/SIFT (*Adzhubei et al., 2010*; *Ng and Henikoff, 2003*) coding region database, the ENCODE (*Pazin, 2015*) and RegulomeDB

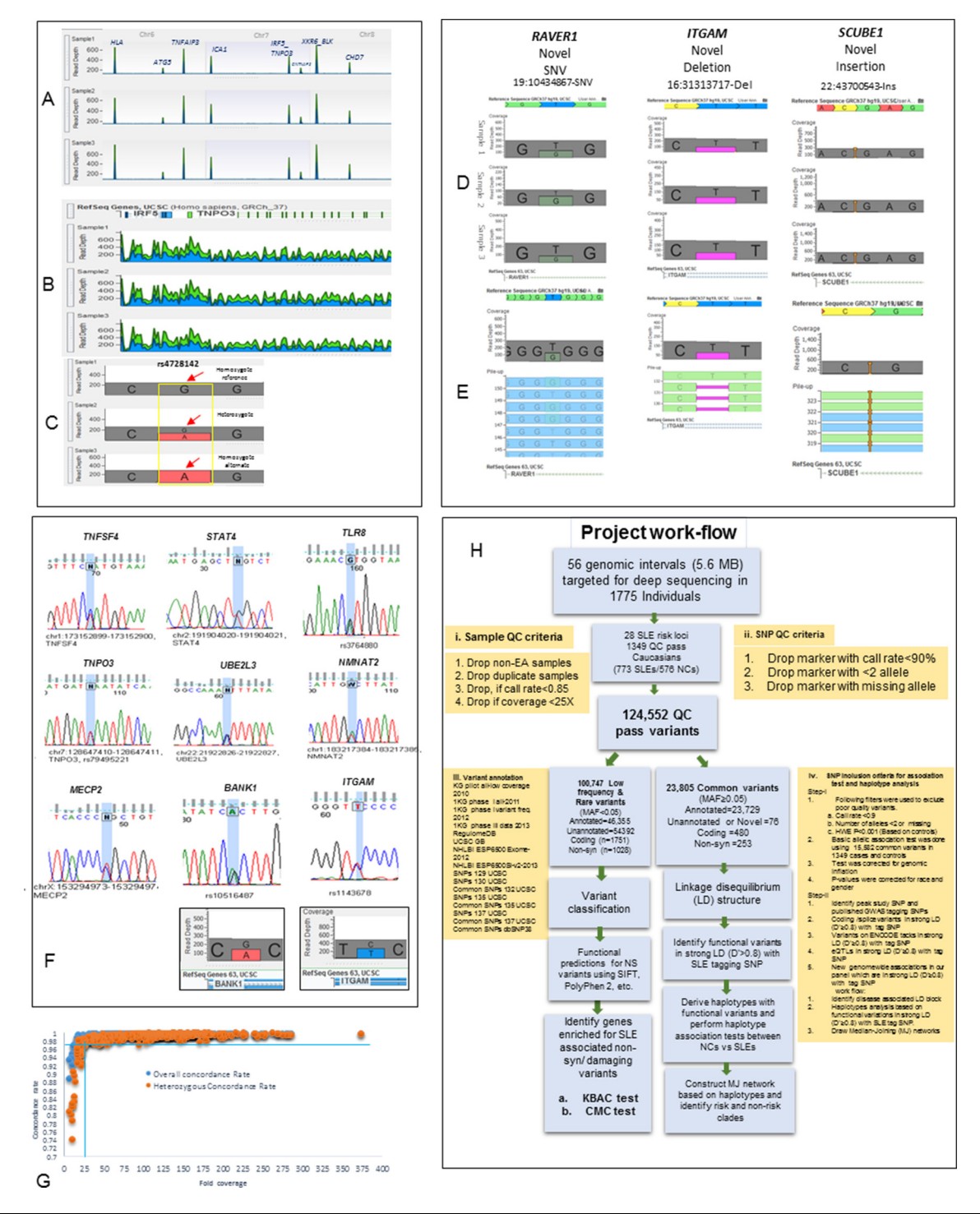

**Figure 1.** Sequencing quality metrics and work flow pipeline. (**A**) Depth of sequence reads across chromosomes 6, 7 and 8 for three samples, illustrating enrichment efficiency for targeted regions. (**B**) Zoom in read depth analysis of *IRF5-TNPO3* gene region (~228 Kb) for three different samples. (**C**) Genotype calls for a SNP in *IRF5* illustrating read depth across a typical variant position. (**D**) Examples of data used to genotype a novel SNV in *RAVER1*, a novel deletion in *ITGAM* and a novel insertion in *SCUBE1* gene. (**E**) The distribution of variant calls in forward and reverse sequencing reads. (**F**) About 35 SNPs from various targeted genes were confirmed by Sanger sequencing. Sanger sequencing results were further validated by calculating read depths for reference and alternate alleles in heterozygous samples, as shown for *ITGAM* and *BANK1*. (**G**) This figure compares fold coverage versus SNP concordance rate for a subset of samples that were both sequenced and genotyped with the Immunochip.
*Figure 1 continued on next page*

*Figure 1 continued*
v1 SNP array. (H) A diagram of the work flow pipeline for bioinformatics analysis of the sequencing data including quantitative information for the number of variants passing filters at each step.

databases (*Boyle et al., 2012*), and several eQTL databases for immune cell lineages (*Fairfax et al., 2014*; *Raj et al., 2014*; *Westra et al., 2013*). The specific technologies, bioinformatics algorithms, and quality assessments used to generate these data are discussed in Materials and Methods and relevant data are provided in *Supplementary file 1B* and *Figures 1A–H*. Overall, these sequence analyses identified 70,070 previously annotated variations and 54,482 novel or unannotated variations within the EA cohort. Functional annotation defined about 40% of the variants in the dataset as regulatory, based on their inclusion in eQTL datasets or their localization into ENCODE-defined regulatory segments (*Figures 2B,C*).

## Association analysis of common variants with SLE

As shown in *Figure 3A*, multiple variants in 26 of the 28 risk loci were strongly associated with susceptibility to SLE, with seven loci reaching genome-wide significance ($p \leq 5 \times 10^{-8}$), ten reaching suggestive significance ($p \leq 5 \times 10^{-5}$), and nine reaching confirmatory significance ($p \leq 10^{-3}$) (tabulated in *Supplementary file 1D*). We also replicated associations previously reported in SLE GWAS for 36 SNPs at ten loci (*Supplementary file 1E*), although the bulk of the strongest associations detected in the sequence dataset were variants that were not previously reported to be associated with SLE. As tabulated in *Supplementary file 1F*, 673 variants in the sequencing data set exhibited similar or stronger associations with disease than published tagging SNPs, and 345 of these were categorized as functional. This is presented in *Figure 3B*, in which functional variants are shown as yellow points, variants with no functional annotations in blue, and previously identified tagging SNPs in red. Zoom in Manhattan plots of *TNFAIP3* and *ITGAM* are also shown. These results show that multiple, new variants had the strongest disease-associations in 27 of the 28 risk loci and that 14 of the peak variants are annotated as functional.

Sixteen risk loci were selected for more detailed analyses, based predominantly on the presence of multiple variations showing strong associations with disease. *Table 1* provides association statistics, identifies the strongest associated variants, and tabulates the coding and non-coding functional variants in tight LD with the peak signal(s) in each locus. As shown, conditional analyses identified four risk loci with multiple, independent signals. This indicates that *NMNAT2-SMG7*, *TNFSF4*, *HLA-D*, and *XKR6* each contained two or more LD blocks with potentially regulatory variants which might be contributing to disease susceptibility independently. In this regard, we attribute regulatory characteristics to these variants based on published studies from the ENCODE consortium and other research groups (see *Supplementary file 1F* for details). Additional studies will be required to confirm these regulatory properties and delineate the precise mechanisms impacting disease-relevant mechanisms. As shown, 1206 functionally annotated variants were in tight LD (D'>0.8) with the 21 peak risk signals and all but 7 of these were non-coding, regulatory variants. These results demonstrate that multiple functional variations are in tight LD with the peak disease associated signal in every risk locus.

## Haplotype analysis of functional variants in tight LD with peak tagging SNPs

The strategy utilized to assess the association of functional variations with disease is outlined in *Figure 1H* (iv) and illustrated for the STAT4 risk locus in *Figure 4*. As shown in *Figure 4A* and tabulated in *Supplementary file 1B*, targeted sequencing of the 104.2 kb STAT4 risk locus produced an average of 100.17-fold coverage and identified 2273 high quality variants. The LD structure of this region was assessed using 104 common markers (MAF>0.1). As shown, two distinct LD blocks were identified and the ~68 Kb LD block that encompasses the 3' portion of STAT4 contained the SLE disease-tagging SNPs. *Figure 4B* plots the disease association of all of the common variants within this LD block and *Figure 4C* demonstrates that conditioning with the strongest SLE tagging SNP (rs12612769) accounts for all of the disease association within STAT4. These results indicate that functional variations in tight LD with rs12612769 are responsible for the disease-associated

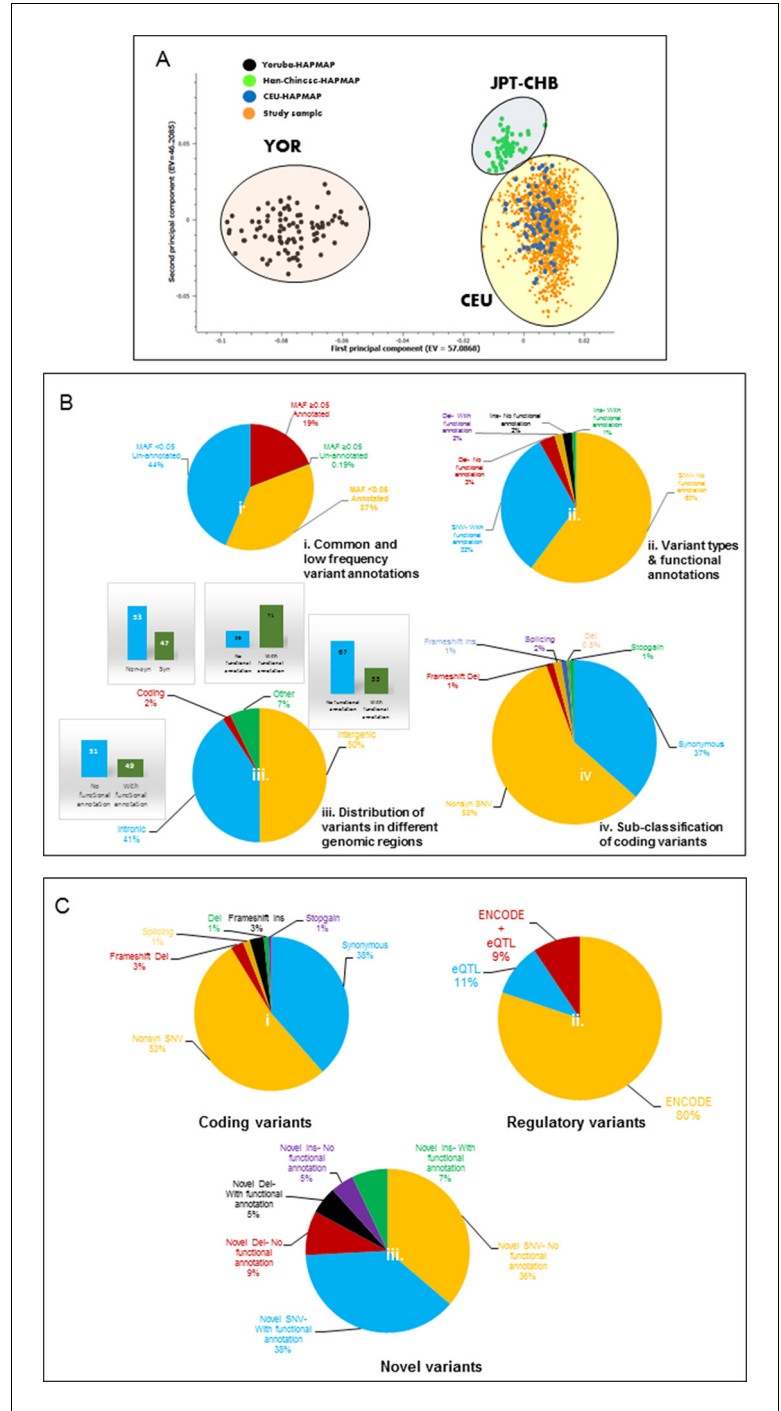

**Figure 2.** Principal component analysis (PCA) and variant summary. (A) Principal component analysis (PCA), showing clustering of study cohort (orange points) with the CEU (blue points) HAPMAP reference group for Caucasians. (B) (i) Pie chart showing percentages of annotated and unannotated variants in common (MAF≥0.05) and low frequency (MAF<0.05) categories. (B) (ii) Pie chart showing percentages of potentially functional single nucleotide variants (SNVs) and structural variants (InDels) defined by ENCODE and eQTL data. (B) (iii) Pie chart showing the distribution of variants in various genomic regions and percentage of potential functional variants in each. (B) (iv) Pie chart showing classification of coding variants into various sub-categories. (C) (i) Pie chart showing classification of common frequency coding/splice variants. (C) (ii) Pie chart showing percentages of ENCODE and/ or eQTL defined potentially functional common regulatory variants. (C) (iii) Pie chart showing the percentages of un-annotated or novel SNVs and InDels with potentially functional annotations.

endophenotypes of the STAT4 locus. *Figure 4D* demonstrates that seven functional variants are in strong LD (D'>0.8) with rs12612769 (strongest tagging SNP in this analysis) and rs7574865 (strongest tagging SNP from literature). *Figure 4E* presents 4 prevalent (frequency > 0.05) haplotypes formed by these functional variants, which in sum account for >90% of the chromosomes found among the 1349 EA samples. As shown, HAP2 is strongly associated with susceptibility to SLE (6.88E-08) and HAP1 is associated with protection (6.00E-04).

*Figure 4F* presents the patterns of sequence divergence that distinguish these haplotypes, utilizing the median neighbor joining (MJ) algorithm (*Bandelt et al., 2000*). MJ analysis is a phylogenetic algorithm that models sequence-based allelic divergence of haplotypes within species. For MJ diagrams, the spheres (termed nodes) represent individual haplotypes in the network and their size is proportional to their frequency. The pie charts overlaid on each node represent the relative frequency of that haplotype in cases (red) and controls (white). The individual SNPs that distinguish each node are listed along the line that connects them and the length of the line is roughly proportional to the number of SNPs that distinguish the haplotypes. The network is progressive, such that the two nodes at opposite ends of the network are most divergent. In essence, MJ analysis provides a visually informative illustration of the relationships of a set of haplotypes segregating within a population.

Several features of STAT4 polymorphisms within the EA population are apparent from this analysis. First, HAP1, HAP3, and HAP4 form a clade of protective haplotypes (nodes highlighted in blue), all with decreased frequencies in SLE patients. Further, both the peak signal SNP in this analysis (SNP5) and the peak GWAS SNP from the literature (SNP6) together with three functional variants, SNP2, SNP8, and SNP9, distinguish the disease-associated HAP2 (highlighted in red) from the haplotypes in the protective clade. As listed in *Supplementary file 2*, SNP8 (rs10181656) is located within a binding site for the CCCTC-binding factor (CTCF), which is a chromatin insulator that inhibits transcription and plays a role in defining the borders of transcriptional domains. SNP9 (rs7582694) is located within an ENCODE-defined segment containing transcription binding sites for ESR1 (estrogen response elements) and FOS1. Both of these transcription factors are active in multiple tissues and immune cell lineages and both are annotated by ENCODE with strong effect scores and good regulomeDB scores, suggesting that these variations mediate transcriptional endophenotypes in several cell lineages.

Finally, SNP2 (rs11889341) is the most potent of several eQTL variants within the STAT4 risk locus is very strongly associated with SLE susceptibility ($p<4.8 \times 10^{-9}$), and impacts the transcription levels of STAT1 and STAT4 (*Supplementary file 2*). As shown in *Figure 4G*, our eQTL dataset for monocyte-derived macrophages (MDM) identifies a significant increase in baseline STAT1 and STAT4 transcription with the T allele of SNP2, which associates this phenotype with susceptibility to SLE. Several other SNPs distinguishing the protective and risk haplotypes were also associated with STAT1 and/or STAT4 transcription levels in published eQTL databases from ex vivo monocytes or peripheral blood (*Supplementary file 2* and *Figure 4H*). These results indicate that the transcription of both STAT1 and STAT4 are impacted by variants in tight LD with the SLE tag variant and that the disease risk allele is associated with increased transcription of both genes in multiple cell types.

## Multiple SLE associations within the HLA-D region

Sequence analysis of the HLA-D region revealed 15129 common variants (1 variant/29.6 bp) distributed throughout the 448 Kb segment analyzed. These variations occur predominantly in non-coding regions, indicating that the entire HLA-D region is diversified. This result is consistent with previous genomic sequencing analyses of HLA-D that defined 4–5 phylogenetic clades with ancient origins for this segment of HLA-D (*Raymond et al., 2005*). *Figure 5A* presents the LD structure of HLA-D within the EA cohort, based on 8062 common variants (MAF>0.15). Overall, LD in the region is high and the LD structure is very complex, with multiple partial LD associations exhibited between various blocks throughout the region.

More common (15129) than rare (12076) variants were detected in HLA-D, which differs significantly from the roughly 5-fold excess of rare variants that are typically detected throughout the human genome and at other SLE risk loci in our study (*Abecasis et al., 2010*; *2012*; *Hu et al., 2014*) (*Supplementary file 1B*). Previous studies in many species have demonstrated that much of the extant polymorphisms found in MHC class I and class II genes have ancient origins and persist in populations over evolutionary timespans (*McConnell et al., 1988*; *Lawlor et al., 1988*;

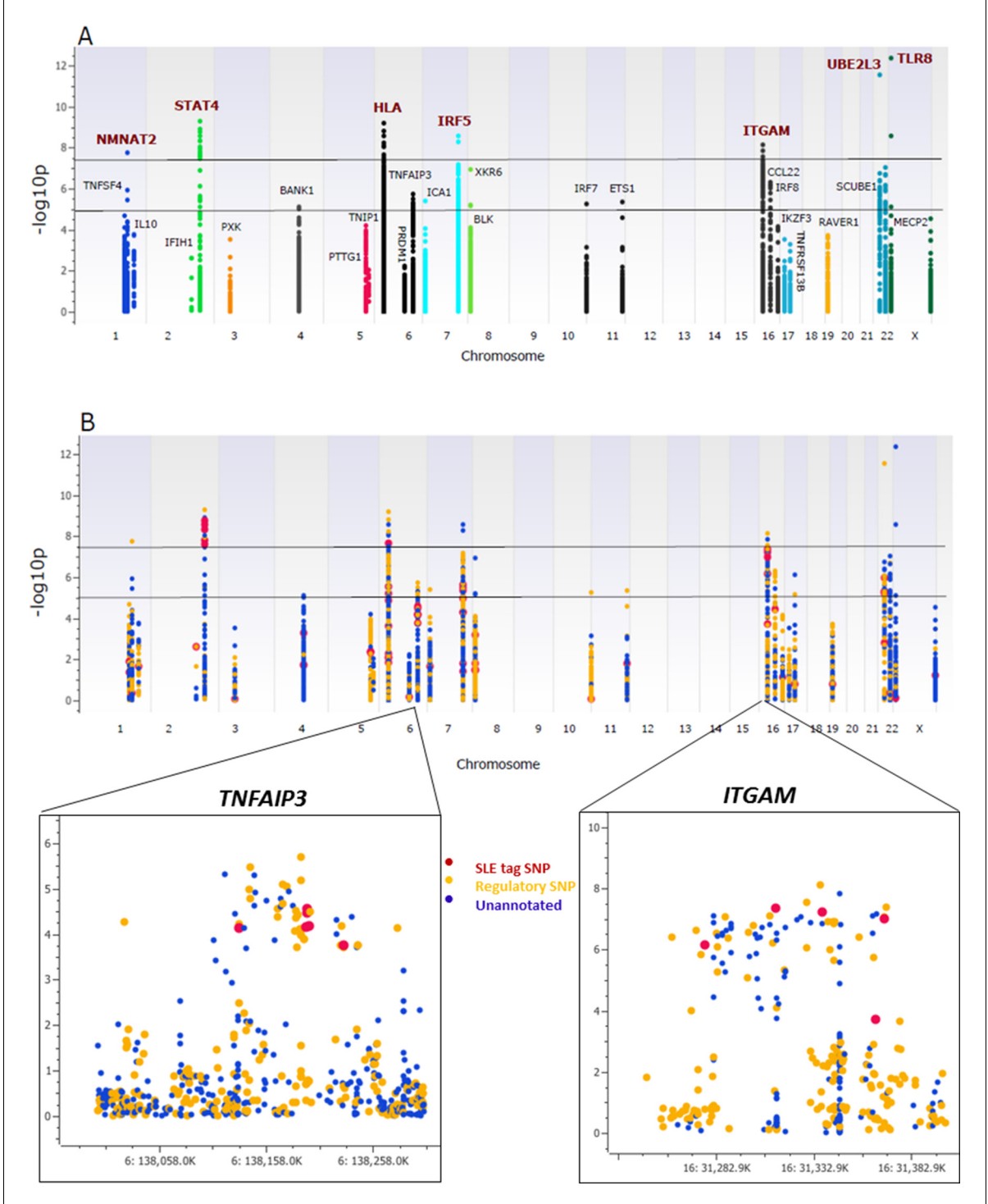

**Figure 3.** Association analysis of sequencing variants from 28 SLE risk loci. (A) Manhattan plot of 15582 common variants (MAF>0.05) plotting –log10 p-value of SLE association (y-axis) versus chromosomal location (x-axis). Horizontal lines mark threshold of significant (p=10⁻⁸) and suggestive (p=10⁻⁵) genome-wide significance threshold. (B) Same Manhattan plot using color coding to identify functional variants (yellow), variants with no current functional annotation (blue), and previously identified SLE GWAS tagging SNPs (red). Zoom in picture of Manhattan plot for *TNFAIP3* and *ITGAM* gene is shown.

**Table 1.** Characteristics of disease associated variants at sixteen SLE risk loci.

| Risk locus | Signal | Peak SNP | Minor allele | Odds ratio (Minor allele) | Allele Freq. (Cases) | Allele Freq. (Controls) | SLE association P-value | SLE associated Annotated variants | Variants in LD with peak SNP (D' >0.8) | | |
| --- | --- | --- | --- | --- | --- | --- | --- | --- | --- | --- | --- |
| | | | | | | | | | Total variants | Total potentially functional variants | Total coding variants |
| STAT4 | 1 | rs12612769 | C | 1.7 | 0.29 | 0.19 | 5E-10 | 52 | 49 | 9 | 0 |
| HLA-D | 1 | rs9271593 (XL9) | C | 1.7 | 0.55 | 0.42 | 7E-10 | 835 | 530 | 398 | 0 |
| | 2 | rs9274678 (DQB1) | G | 2.1 | 0.24 | 0.13 | 6E-09 | 736 | 216 | 69 | 0 |
| | 3 | rs36101847 (DRB1) | T | 0.5 | 0.13 | 0.23 | 8E-09 | 760 | 296 | 126 | 0 |
| ITGAM-ITGAX | 1 | rs41476751 | C | 1.9 | 0.25 | 0.15 | 8E-09 | 153 | 121 | 62 | 3 |
| IRF5_TNPO3 | 1 | rs34350562 | G | 1.8 | 0.23 | 0.14 | 3E-09 | 245 | 189 | 124 | 0 |
| UBE2L3 | 1 | rs181366 | T | 1.5 | 0.27 | 0.20 | 2E-07 | 82 | 79 | 55 | 1 |
| BANK1 | 1 | rs4699260 | T | 0.7 | 0.20 | 0.28 | 9E-06 | 267 | 143 | 29 | 2 |
| TNIP1 | 1 | rs62382335 | A | 1.4 | 0.14 | 0.10 | 6E-05 | 46 | 22 | 16 | 0 |
| TNFAIP3 | 1 | rs57087937 | T | 1.9 | 0.10 | 0.06 | 2E-06 | 69 | 63 | 40 | 1 |
| CCL22-CX3CL1 | 1 | rs223889 | T | 1.5 | 0.34 | 0.27 | 5E-07 | 32 | 25 | 20 | 0 |
| RAVER1-ZGLP1 | 1 | rs35186095 | T | 1.3 | 0.21 | 0.17 | 2E-04 | 43 | 24 | 19 | 0 |
| ICA1 | 1 | rs74787882 | A | 0.7 | 0.06 | 0.09 | 2E-03 | 34 | 10 | 6 | 0 |
| TNFSF4 | 1 | rs1819717 | G | 0.7 | 0.29 | 0.36 | 2E-05 | 73 | 30 | 14 | 0 |
| | 2 | rs4916313 | C | 1.3 | 0.39 | 0.32 | 2E-04 | | 30 | 21 | 0 |
| BLK | 1 | rs7822109 | C | 0.8 | 0.46 | 0.52 | 9E-05 | 97 | 61 | 38 | 0 |
| XKR6 | 1 | rs4840545 | A | 2.0 | 0.13 | 0.07 | 1E-07 | 335 | 51 | 23 | 0 |
| | 2 | rs7000132 | C | 0.9 | 0.42 | 0.46 | 5E-04 | | 178 | 118 | 0 |
| NMNAT2-SMG7 | 1 | rs41272536 | G | 2.9 | 0.11 | 0.05 | 2E-08 | 33 | 8 | 8 | 0 |
| | 2 | rs111487113 | A | 0.6 | 0.13 | 0.18 | 5E-04 | | 17 | 5 | 0 |
| ETS1 | 1 | rs34516251 | A | 0.8 | 0.18 | 0.21 | 7E-03 | 18 | 10 | 6 | 0 |

*Gyllensten and Erlich, 1989*; *Edwards et al., 1997*; *She and Wakeland, 1991*). The preponderance of common variants in this sequence dataset is consistent with an ancient origin of these HLA-D polymorphisms within the human lineage. In addition, the LD structure of these HLA-D variations in our EA cohort is very similar to the LD structure obtained for this segment of HLA-D in the 2504 human genomes in the 1000 genome project (*Figure 5Ai and ii*).

*Figure 5B* plots the association of variants throughout HLA-D with SLE and uses color-coding to distinguish regulatory variants (yellow) from variants with no current functional annotation (blue). As shown, 3786 variations in HLA-D are nominally associated with SLE (p<0.05) and 1797 of these have annotated regulatory functions. Five separate segments within HLA-D contained variants achieving genome-wide significance for association with SLE. The peak SLE association was with rs9271593 (p=6.50E-10), which is one of 687 SLE-associated regulatory variants mapping to the XL9 regulatory component within the intergenic region separating DRB1 and DQA1 (*Majumder et al., 2006*; *Majumder et al., 2008*). This ~50 kb segment is heavily annotated with ENCODE-defined regulatory sequences controlling chromatin structure and/or the binding of specific transcription factors. As shown in *Figure 5Aiii & iv* the XL9 segment of HLA-D is not adequately covered by SNP typing

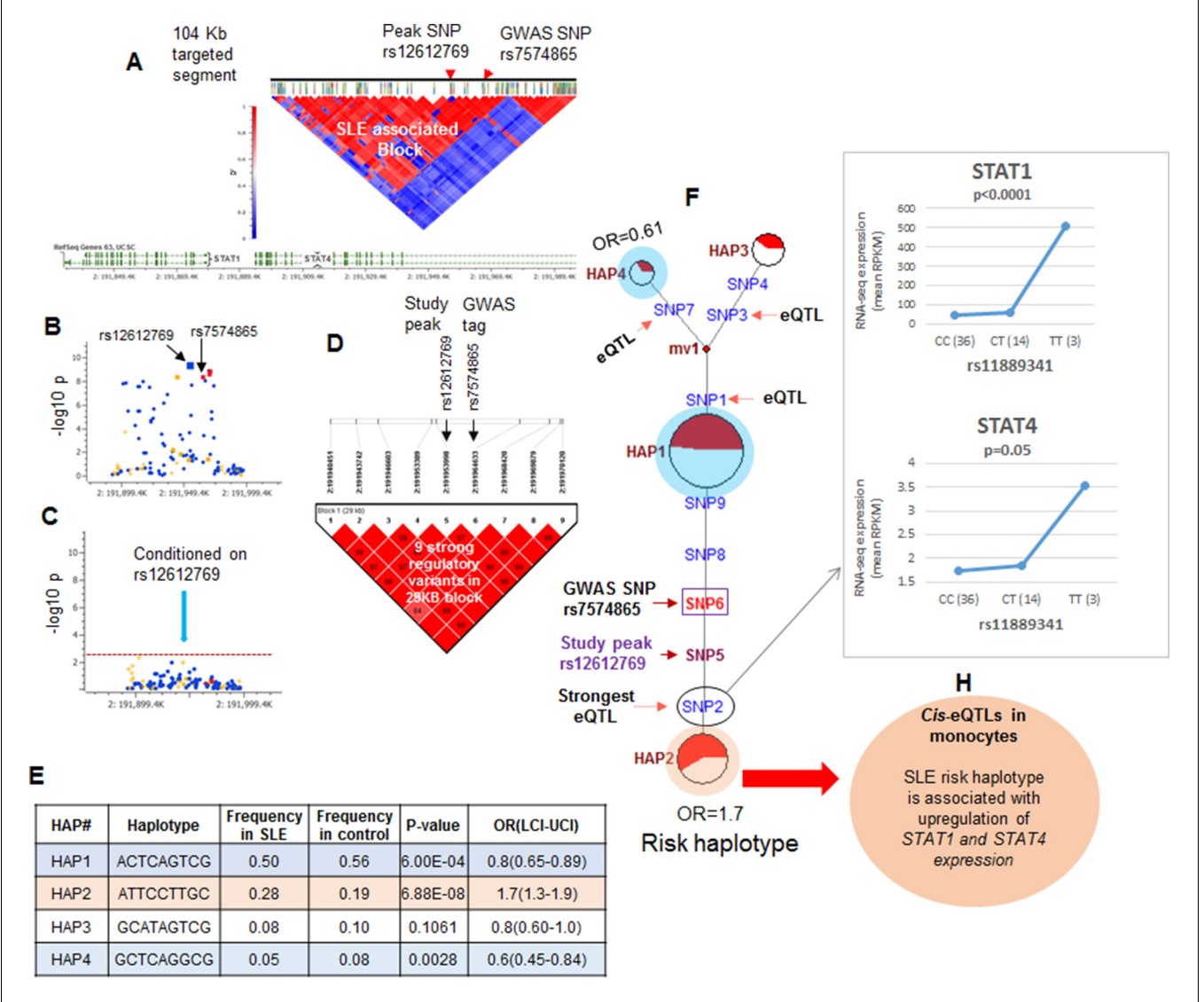

**Figure 4.** LD structure, haplotypes and MJ networks analysis at *STAT4* locus. (A) LD structure of *STAT4* sequenced segment is shown above molecular map of the genomic segment showing *STAT1* and *STAT4* exon structure. The locations of GWAS tagging SNPs are shown above LD plot, which was produced with 104 markers (MAF≥10%) in 1349 Caucasians. (B) Zoom in Manhattan plot showing SLE association levels of individual sequence variants in *STAT4* LD block containing STAT4 tagging SNPs. Yellow points indicate functional variants, blue points indicate un-annotated variants and red points identify GWAS and study peak tagging SNPs. (C) Conditional analysis on peak SNP rs12612769 removes all significant associations with SLE within the LD block. (D) LD block based on nine potentially functional SLE associated variants used for haplotype analysis. (E) Derived haplotypes with SLE association results. (F) Median-joining (MJ) network analysis of *STAT4* haplotypes. Spheres (termed nodes) represent the locations of each haplotype (from table in E) within the network and the size of the node is proportional to the overall frequency of that haplotype in the dataset. Each node is overlaid with a pie chart that reflects the frequency of that haplotype in cases (red) versus controls (white). The lines connecting the nodes are labeled with the variants that distinguish the connected nodes and the length is proportional to the number of variants. Haplotypes with significant (p<0.05) association with SLE are highlighted with red (risk) and blue (non-risk). Study peak SNP, SLE GWAS tag SNP and eQTLs are indicated with arrows, boxes and circles within their locations within the network. (G) Presents cis-eQTL effects observed with SNP2 on *STAT1* and *STAT4* in macrophage RNAseq analysis. (H) Similar eQTL effects observed in published eQTL databases in literature.

arrays such as the Immunochip v.1, possibly accounting for the failure to detect this potent SLE association in previous GWAS analyses. The second strongest SLE association signal was with rs9274678 (p=6.21E-09), which is located in a segment extending 5′ from the DQB1 proximal promoter. The third genome-wide SLE signal was with rs36101847 (p=8.33E-09), which has no functional annotation and is located in close proximity to DRB1 exon 2, which encodes the peptide binding segment of

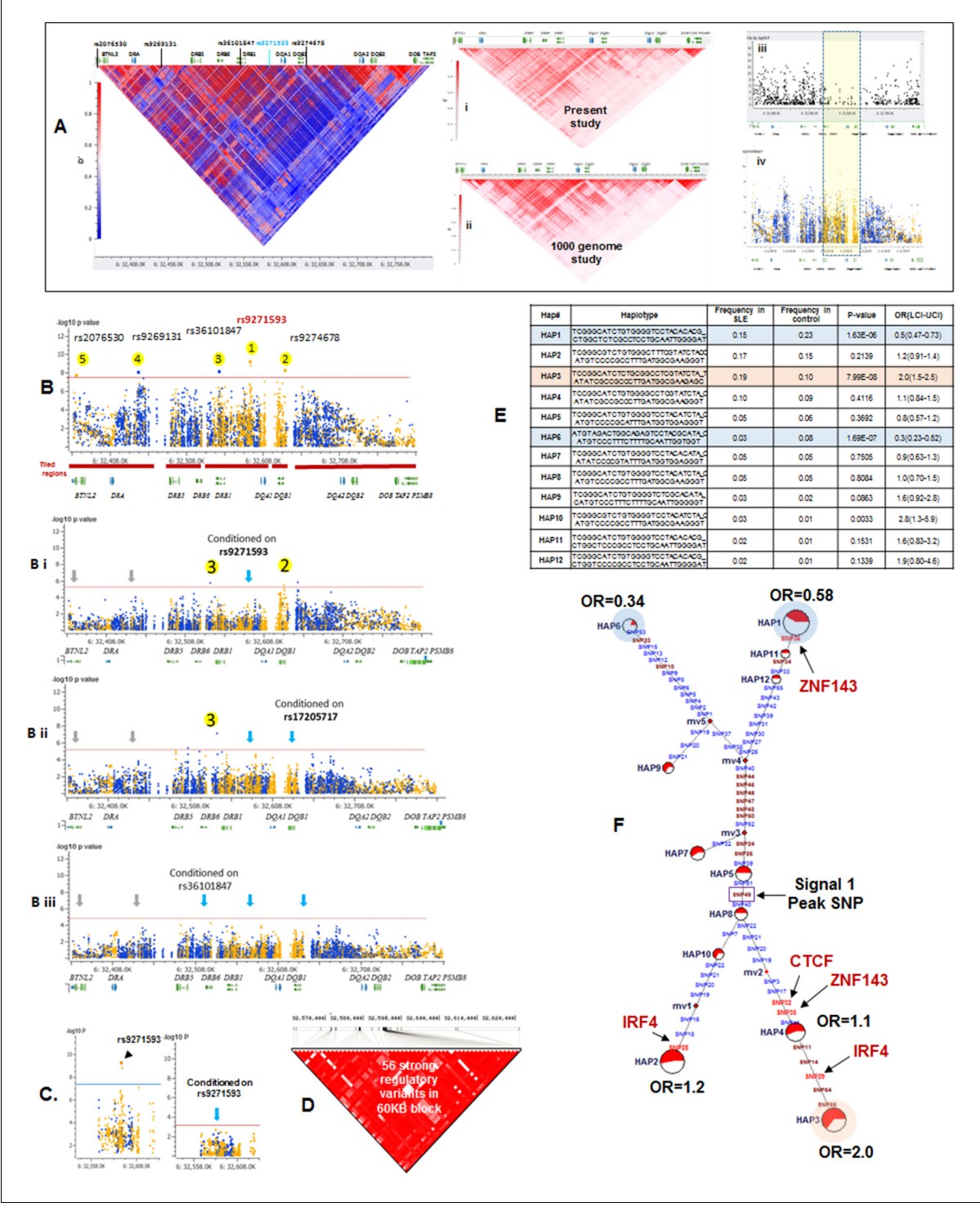

**Figure 5.** LD structure, haplotypes and MJ Network analysis of XL9 region. (**A**) The LD structure of HLA-D region is shown below a molecular map of the region. The locations of the genes and five genome-wide SLE association signals are marked. Peak association signal is coded blue. (**A**) (i) The HLA-D LD structure in 1349 Caucasians from present study assayed with -8062 common (MAF>0.15) variants. (**A**) (ii) The HLA-D LD structure in 2504 samples representing twenty-six cohorts from the world population. Data obtained by analysis of the1000 Genome project datasets using the same -8062 variants analyzed in A (i). (**A**) (iii) SNP content of Immunochip v.1 across HLA-D region. (**A**) (iv) High quality common variant calls in this region from targeted sequencing in this study. Highlighted area boxes the XL9 through DQB1 5' segment regulatory region. Yellow points indicate potentially functional variants and blue points indicate un-annotated sequencing variants. (**B**) Zoom Manhattan plot of all common (MAF>0.05) variants using color coding to identify functional (yellow) and non-annotated (blue) variants. The locations of peak association signals are marked. A molecular map of the

*Figure 5 continued on next page*

*Figure 5 continued*

region and the tiled regions for targeted sequencing are identified at the bottom. Gaps reflect the locations of long stretches of highly repetitive regions that cannot be assembled. (**B**) (**i**) The residual association level after conditioning on peak signal 1 in XL9. (**B**) (**ii**) Residual association level after conditioning on both signal 1 (XL9) and signal 2 (DQB1 5' segment). (**B**) (**iii**) No significant associations remain after conditioning on signal 1 (XL9), signal 2 (DQB1 5' segment), & signal 3 (DRB1). Yellow points identify potentially functional variants and blue points indicate un-annotated variants. (**C**) Conditional analysis on peak SNP rs9271593 (XL9 signal) showing that all significantly associated variants are in tight LD. (**D**) A 60KB LD block generated with 56 variants from XL9 region with strong regulatory scores and association with SLE. (**E**) Twelve haplotypes generated with HAPLOVIEW using the 56 regulatory variants. Frequencies in cases and controls, association statistics, and odds ratios are provided. Protective (blue) and risk (red) haplotypes are highlighted. (**F**) Median neighbor-joining (MJ) network produced as described in the text. Annotation is the same as presented in legend for *Figure 2*. Variants that disrupt binding sites of CTCF, ZNF143, and IRF4 are labeled.

DRB1. The fourth signal is rs9269131 (p=9.92E-09), which has no functional annotation and is in close proximity to DRA1. Finally, the fifth signal is rs2076530 (p=2.21E-08), which is a regulatory SNP in proximity to BTNL2 that has previously been associated with sarcoidosis and autoimmunity (*Hofmann et al., 2013*).

The independence of these five SLE association signals was assessed by conditional analysis, beginning with the peak SNP (XL9 region, rs9271593) and using a forward stepwise regression method (*Figure 5B*). As shown in *Figure 5Bi*, conditioning with rs9271593 removed the associations of signals 4 and 5 with SLE, consistent with the LD associations revealed in *Figure 5A*. These results indicate that the DRA1 and BTNL2 disease association signals are in strong LD with variants in the XL9 region. However, the DQB1 promoter and DRB1 signals, although significantly diminished, were still significantly associated with disease after removal of the peak XL9 signal, indicating that these signals were somewhat independent. Removal of the XL9 and DQB1 promoter signals left only the DRB1 signal with marginally significant SLE association and removal of all three signals removed all significant association of HLA-D with disease (*Figure 5Bii & iii*). Thus, these conditional analyses identified three independent disease associations, each localized to important regulatory or coding elements within HLA-D. Their basic properties and SLE association characteristics are summarized in *Table 1*.

## XL9 polymorphisms mediate quantitative variations in HLA-DR and – DQ transcription

As shown in *Figure 5C*, conditioning on the XL9 signal (rs9271593) completely removed the SLE association of variants within the DRB1 to DQA1 intergenic segment, indicating that rs9271593 tags the XL9 functional haplotype responsible for this disease signal. The content of regulatory variants in strong LD (D' >0.8) with the XL9 signal was very high (*Table 1*) and consequently detailed haplotype analysis was focused on variants with strong ENCODE functional effect scores (>500 for at least one transcription factor binding site), eQTL effects, and associations with SLE. This identified 56 functional variants that formed 12 haplotypes of which HAP3 was strongly associated with SLE susceptibility (OR 2.0, p<7.04 E-08) while HAP1 (OR 0.58, p<1.63 E -06) and HAP6 (OR 0.34, p<1.69 E-07) were protective (*Figure 5D,E*, *Supplementary file 2*). As shown in *Figure 5F*, MJ analysis found that the protective (shaded in blue) and risk (shaded in red) haplotypes form separate clades at opposite ends of the network, indicating that these two extremes in disease association are also extremes in the divergence of regulatory variations.

*Figure 6A* overlays the locations of the three peak disease signals in HLA-D with the multitude of regulatory elements located within the ~130 Kb segment spanning *HLA-DRB1* through *HLA-DQB1*. This genomic segment contains 5 separate regions with dense arrays of sequence elements that regulate chromatin structure (histone marks, DNAse I clusters) and transcription factor binding. As shown in the top tract within *Figure 6A*, more than 150 variations within this small genomic segment are eQTLs that have been shown to impact the transcription of 72 genes in various immune cell lineages (*Supplementary file 2*). Our own RNA-SEQ-based eQTL dataset for MDM associates many of these variations with quantitative variations in the transcription of DRB1, DQA1, and DQB1. All of the MDM eQTL variants impacting HLA-DR and DQ expression are associated with variants in XL9 (*Figure 6D*).

XL9 contains binding sites for multiple factors affecting chromatin structure (CTCF, ZNF143) and has been shown to assemble HLA-DR and HLA–DQ proximal promoters into a transcriptional

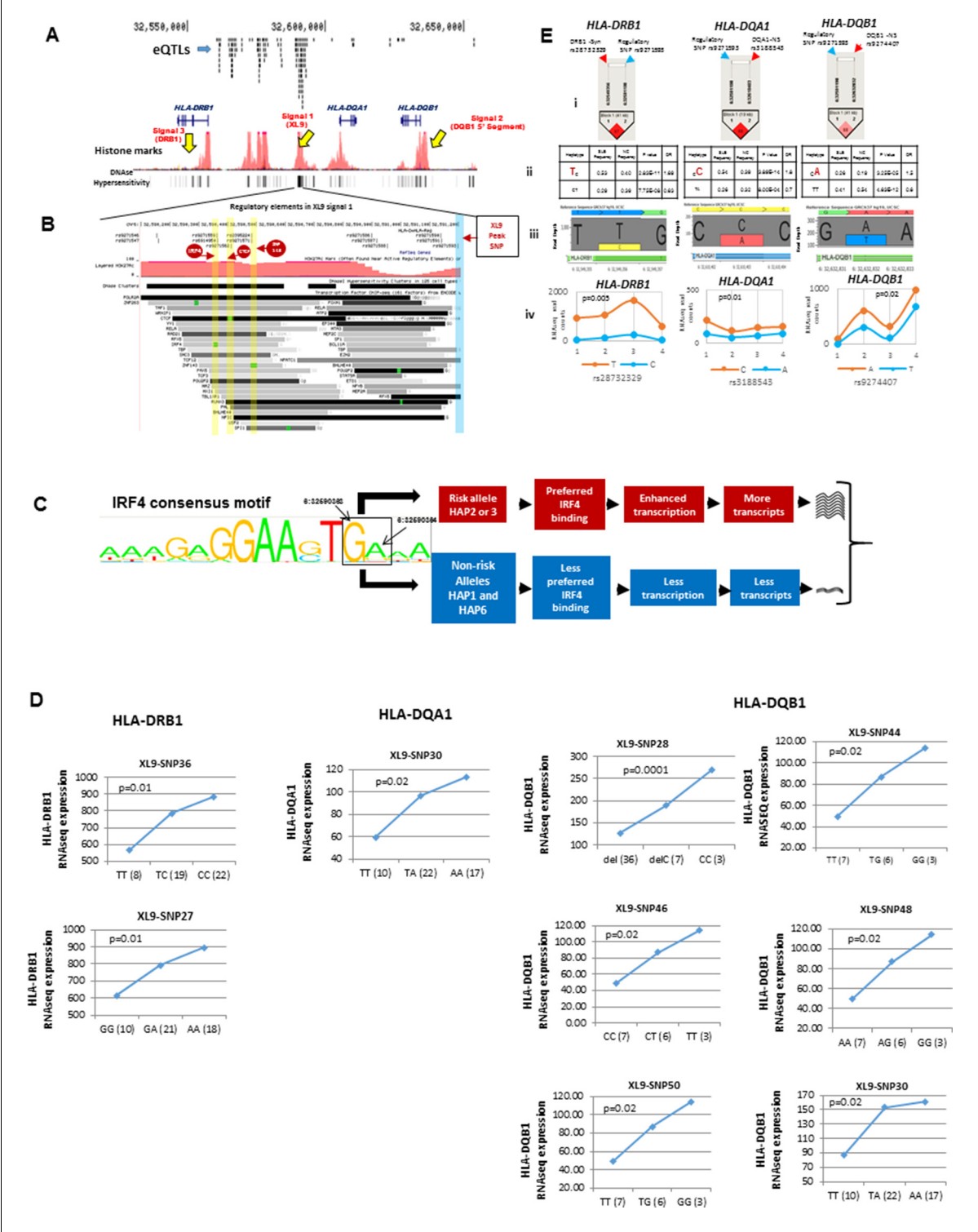

**Figure 6.** Chromatin architecture and transcriptional regulation at SLE associated XL9 region. (**A**) A snap shot of the ~140 Kb DRB1-DQB1 segment that contains three genome-wide association signals for SLE. The locations of HLA class II genes and the peak signals are marked. The locations of some of the more than 750 eQTLs variants mapped into this region are overlaid onto ENCODE defined regulatory elements (Histone marks and DNA hyper sensitivity clusters). (**B**) A snap shot of a ~1 Kb segment in the center of the XL9 that contains 13 of the 56 strong regulatory variants that constitutes the XL9 haplotype. The positions of the canonical protein binding motifs of CTCF, IRF4 and ZNF143 highlighted in yellow and the peak XL9 SNP highlighted in blue. The locations of about 30 binding sites for transcription factor that are located within this same region and are also impacted genetic variation are also listed. (**C**) The consensus sequence for IRF4 binding in XL9 is shown with the locations of the two nucleotide variants boxed

*Figure 6 continued on next page*

Raj *et al*. eLife 2016;5:e12089. DOI: 10.7554/eLife.12089

*Figure 6 continued*

and marked. The consensus sequence for IRF4 binding (GA) are the alleles present in XL9 risk haplotypes. The alternative alleles for these two nucleotides, which are much less frequent in IRF4 binding motifs, are in protective haplotypes. The red and blue highlighted paths describe the predicted effects of these variations on IRF4-mediated transcription of HLA-DR and HLA-DQ, with risk haplotypes highlighted in red and protective haplotypes highlighted in blue. (D) shows cis eQTL effects observed with SLE associated XL9 region regulatory variants. SNPs were found to impact the expression level of *HLA-DRB1, HLA-DQA1* and *HLA-DQB1* gene in monocyte derived macrophages (MDMs). In each plot, x-axis shows three genotypes of a given eQTL SNP and y-axis shows RNAseq expression values in RPKM. SNP numbers correspond to XL9 variants in *Figure 5F*. (E) Part i shows LD between peak regulatory SNP and a coding SNP in HLA-DRB1, DQA1 and DQB1. Part ii highlights the SLE associated coding allele sequence and shows the association statistics on peak regulatory and coding SNP haplotype for above three genes. Part iii shows the allelic bias in transcription in DRB1, DQA1 and DQB1 gene in human macrophages, demonstrated in terms of significantly different number of RNA sequencing reads for SLE risk and non-risk allele. Part iv shows the transcriptional bias between risk and protective alleles for HLA class II genes in four heterozygous human donors for these IRF4 variants.

complex that facilitates coordinate, tissue-specific transcription (*Bailey et al., 2015*; *Xi et al., 2007*; *Liu et al., 2008*; *Whitfield et al., 2012*). This assembly is mediated by interactions of the transcriptional insulator protein CTCF with cohesion and other chromatin regulatory components such as ZNF143. Current data supports the idea that the XL9 transcriptional complex contains multiple regulatory domains that interact with a variety of transcription factors to control the coordinated expression of HLA-DR and -DQ in lymphoid, myeloid, and thymic epithelial cell lineages. For example, recent studies by Singh and co-workers have demonstrated that the level of transcription of HLA class II molecules in dendritic cells is strongly controlled by the transcription factor IRF4, which is one of several transcription factor binding sites located within XL9 (*Vander Lugt et al., 2014*). Notably, IRF4 up-regulation of MHC class II molecules was shown by these investigators to be strongly associated with disease severity in a murine model of experimental autoimmune encephalomyelitis (EAE).

*Figure 6B* presents a fine map of variations occurring within the regulatory sequence elements in a 1 Kb segment in the center of XL9. This segment contains more than 30 transcription and chromatin configuration factor binding sites. Thirteen of the fifty-six potent regulatory variants in the XL9 haplotypes are located within this small segment. Five of these variants are located in the consensus binding sites for CTCF, ZNF143, or IRF4 and are predicted to impact their binding properties (*Figure 6B*). As shown in the MJ network in *Figure 5F*, four of the five motif variants occur within the SLE risk clade and differ between the protective and risk clades. Interestingly, one of the IRF4 variants occurs in the final link of the risk clade leading to HAP3 (strongest SLE association), while the second IRF4 variant is in the final branch leading to risk HAP2.

*Figure 6C* diagrams the nucleotide changes in the binding motif of IRF4 caused by the two XL9 IRF4 variants, both of which are predicted to strongly impact the binding of IRF4. For both of these variants, alleles carrying the unmodified IRF4 consensus binding sequence are associated with the risk haplotypes, while those with nucleotides that are less common to the consensus motif are present in all of the non-risk haplotypes (*Figure 5F*, *Figure 6C*, and *Supplementary file 2*). As diagrammed in *Figure 6C*, this suggests that the transcription of HLA-DR and -DQ should be increased in individuals carrying risk HAP2 and HAP3 due to increased IRF4 binding to XL9 regulatory domains. As shown in *Figure 6D*, the HAP2 and HAP3 alleles of variants within this short segment, including an IRF4 motif variant, are strongly associated with increased transcription of HLA-DRB1, HLA-DQA1, and HLA-DQB1 in our eQTL panel of MDM cell cultures and in datasets from the literature (*Supplementary file 2*).

The crucial role of the XL9 region in the chromatin configuration of the HLA-DR and DQ transcriptional complex suggests that XL9 variations should impact the transcription of both DR and DQ in a cis-active, chromosome-specific manner. To test this, we measured allele-specific transcription in four individuals that are heterozygous for XL9 alleles within our MDM eQTL panel. Each of these individuals carries a HAP3 XL9 risk haplotype together with either a HAP1 or HAP6 XL9 protective haplotype. We assessed the allelic bias of transcription for DRB1, DQA1, and DQB1 in these heterozygotes utilizing coding region SNPs in tight LD with XL9 regulatory variants. As shown in *Figure 6E*, coding region variants linked with XL9 risk associated regulatory variants are present in a significantly higher proportion of RNA-SEQ reads than coding variants linked to protective alleles. The allelic bias in RNA-SEQ reads for HLA-DRB1, HLA-DQA1, and HLA-DQB1 is presented for a representative coding SNP in *Figure 6E* (iii). This bias in SNP read depth was a consistent feature for

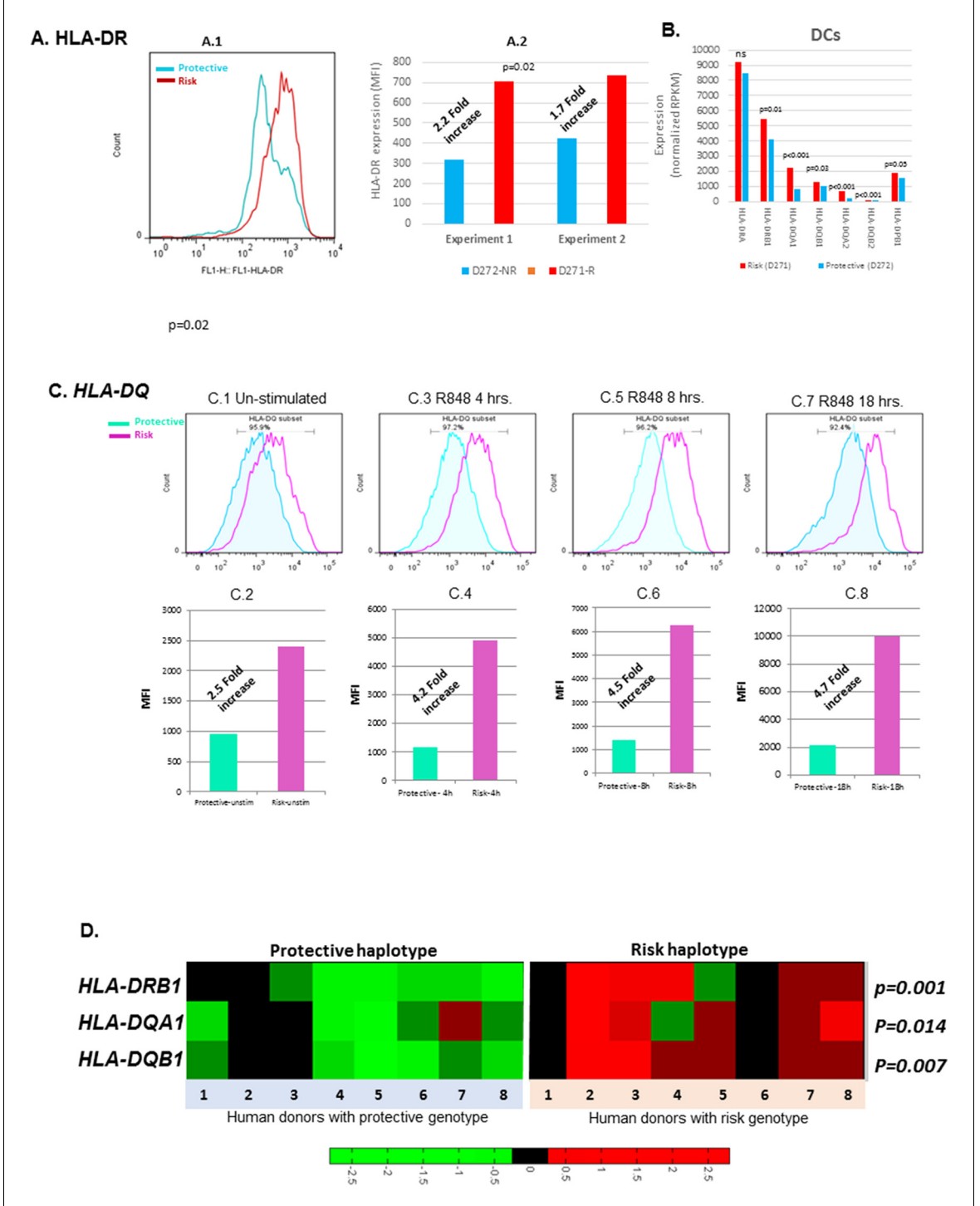

**Figure 7.** Cell surface expression of HLA-CLASS II genes. (**A.1**) Monocyte-derived dendritic cell (MDDC) surface expression of HLA-DR in a culture produced from a homozygote for protective (blue) and homozygote for risk (red) HLA-D haplotypes. This experiment was repeated in same donors. (**A.1**) shows flow data. (**A.2**) shows the MFIs from repeated experiments. p-value shown in (**A.2**) was calculated on mean MFIs from two experiments. (**B**) shows normalized RNAseq expression on HLA-class II genes in dendritic cells on same donors presented in (**A**). (**C.1–C.8**) shows HLA-DQ surface expression on MDDC cultures from a homozygote for protective (blue) and heterozygote for risk (red) HLA-D haplotype. Flow data and respective MFIs are shown on MDDCs at steady state (**C.1** and **C.2**), at 4 hr (**C.3** and **C.4**), 8 hr (**C.5** and **C.6**) and 18 hr (**C.7** and **C.8**) after stimulation with TLR7/8 ligands. (**D**) heatmap on RNAseq data on lymphoblastoid cell line (LCL) from 1000 genome project compare expression level of HLA-class II genes between individuals homozygous for HLA-D protective and risk haplotype.

SNP variants throughout exon 2 and exon 3 for all three genes in all four heterozygous individuals [*Figure 6E(iv)* ]. Taken together, these experiments demonstrate that XL9 regulatory variations modulate the level of transcription of HLA-DR and HLA-DQ in a chromosome-specific manner.

The functional implications of increased transcription of HLA-D by SLE risk associated XL9 alleles are contingent upon their impact on the surface expression levels of HLA-DR and HLA-DQ molecules on immune cell lineages. To test this, quantitative flow cytometry was performed on monocyte-derived dendritic cell cultures (MDDC) derived from the PBMC of individuals with specific XL9 haplotypes. As shown in *Figure 7A.1 and A.2*, HLA-DR surface expression is roughly 2.5-fold higher on unstimulated MDDC from a HAP3 (risk) homozygote in comparison to a HAP1 (protective) homozygote. As shown in *Figure 7A.2*, this statistically significant variation in surface expression was fully reproducible. RNA-SEQ analyses of these same MDDC cultures confirmed the increase transcription of HLA-D genes by the XL9 HAP3 donor (*Figure 7B*). Similarly, the surface expression of HLA-DQ on MDDC from a risk/protective heterozygote is greater than the expression levels of those from a homozygote for a protective haplotype (*Figure 7C.1–C.2*). This increased surface expression of HLA-DQ is maintained on dendritic cells following activation with the TLR7/8 ligand R848 in a time course over 18 hours, indicating that HLA-D risk haplotypes drive higher levels of HLA class II molecule surface expression during TLR activation and dendritic cell maturation (*Figure 7C.3–C.8*). Taken together, these findings indicate that variations in the XL9 regulatory region modify chromatin structure and transcription factor binding, leading to a significant increase in the surface expression of HLA class II in the dendritic cell lineage of individuals expressing SLE risk alleles of HLA-D. Finally, the transcription of several genes within the HLA complex are strongly upregulated in lymphoblastoid cell lines from risk versus protective XL9 haplotype homozygotes in the 1000 genome RNA-SEQ lymphoblastoid cell line (*Lappalainen et al., 2013*) data set (*Figure 7D*). These results, obtained via an identical analysis of a public dataset, are an independent replicate of our findings of increased expression of important HLA genes in individuals carrying HLA-D haplotypes associated with SLE.

## Regulatory effects and disease associations of composite HLA-D haplotypes

Detailed analyses of the SLE signal within the segment 5' of HLA-DQ (signal 2) revealed tight LD with twenty eight regulatory variants that are distributed through a ~40 KB segment extending 5' from the DQB1 transcription start site. The properties of these regulatory variants are provided in *Supplementary file 2* and conditioning and MJ analyses are provided in *Figure 8Ai-vi*. Interestingly, none of the regulatory variants are correlated with eQTL effects that impact DQB1, although they are associated with transcriptional effects on other genes within the antigen processing pathway of HLA, such as PSMB8, PSMB9, TAP1, TAP2, DQA2, and DQB2. Among the variants in tight LD, seven have strong ($\geq$900) effect scores from ENCODE and 4 of these are also scored within the 1 or 2 category in the RegulomeDB database, making it likely that they impact transcription factor binding and transcription (*Supplementary file 2*). ENCODE predicts that these variants will impact the binding of many transcription factors including POLR2A, RELA, BATF, RUNX3, TBP, TAF1, PAX5, RFX5, EP300, and NFIC. The twenty eight variants formed seven haplotypes of which HAP4 was protective (OR 0.3, p<3.88 E-09) and HAP2 condoned risk (OR 1.9, p<1.11 E-06). As shown, two of the strongest regulatory variants and all of the variants showing strong associations with SLE are located on the final MJ branch leading to the risk clade. These results indicate that the most potent regulatory variants identified by ENCODE and associated with the increased transcription of multiple components of the antigen presentation pathway (APP), are all associated with increased risk for SLE.

The DRB1 signal (signal 3) is also associated with twenty eight regulatory variations; however, their predicted functional properties are weaker than those found in the XL9 or DQB1 signals (*Supplementary file 2*). The DRB1 regulatory variants are distributed in a region extending from the peak SNP through DRB1 and about 10 Kb 5' from the DRB1 start site towards the XL9 regulatory region. As shown in *Figure 8Biv*, they form 8 haplotypes of which HAP6 is protective (OR 0.4, p=4.58E-05) and HAP1 is risk (OR 1.6, p=1.50E-06). The DRB1 regulatory variants do not have strong ENCODE or RegulomeDB scores and do not contain eQTL associations with DRB1 expression. However, associations with DRB5, DQA1, and DQB1 transcription have been reported for variants in the DRB1 regulatory haplotypes. Also, although the DRB1 peak variant is strongly associated with SLE, only 2 of the regulatory variants in tight LD with this peak variant are strongly associated with SLE (*Figure 8Bv*) and those variants were not included in the MJ network branch proximal to

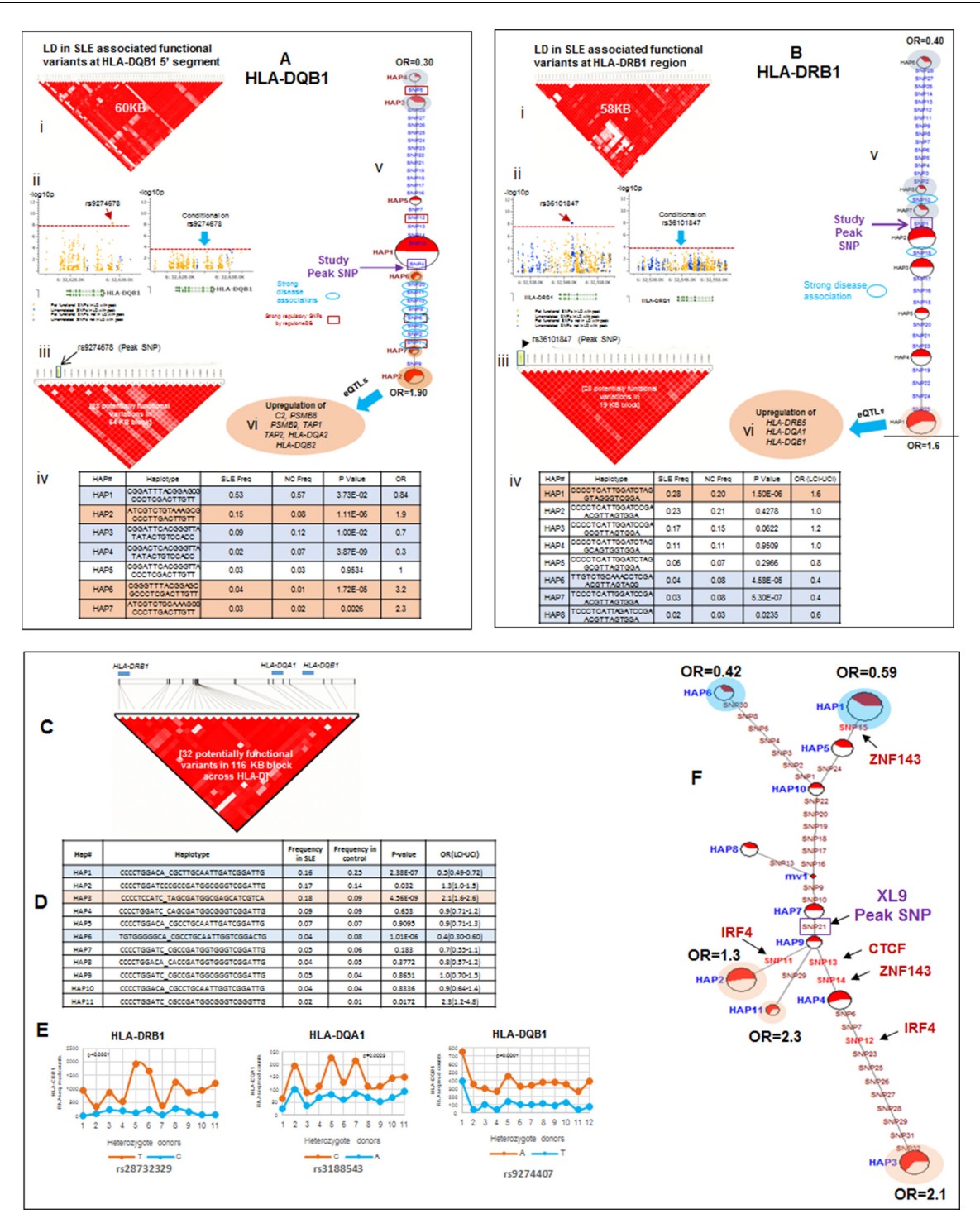

**Figure 8.** LD structure, haplotypes and MJ network analysis in *HLA-DQB1* and *HLA-DRB1* region. (**A**) (i) LD structure at *HLA-DQB1* 5' region generated with 68 common (MAF≥10%) potentially functional variants in 1349 samples. (**A**) (ii) Zoom Manhattan plot showing SLE variant association levels and conditional analysis on peak SNP rs9274678. (**A**) (iii) LD block structure of 28 potentially functional SLE associated SNPs which are used for downstream haplotype analysis. (**A**) (iv) Haploview generated seven haplotypes from these 28 functional variants. Frequencies in cases and controls and association statistics are provided. Risk (red) and protective (blue) haplotypes are color highlighted. (**A**) (v) MJ networks analysis to illustrate divergence of risk and protective regulatory haplotypes. (**A**) (vi) eQTL variations from public databases for variants in strongest risk haplotype. (**B**) (i) LD structure at HLA-DRB1 region generated with 66 common (MAF≥10%) potentially functional variants in 1349 samples. (**B**) (ii) Zoom Manhattan plot showing SLE variant association levels and conditional analysis on peak SNP rs36101847. (**B**) (iii) LD block structure of 28 potentially functional SLE associated SNPs which

*Figure 8 continued on next page*

*Figure 8 continued*

are used for downstream haplotype analysis. (B) (iv) Haploview generated eight haplotypes from these 28 functional variants. Frequencies in cases and controls and association statistics are provided. Risk (red) and protective (blue) haplotypes are color highlighted. (B) (v) MJ networks analysis to illustrate divergence of risk and protective regulatory haplotypes. (B) (vi) eQTL variations from public databases for variants in strongest risk haplotype. Panel (C) 116 kb LD block generated with 32 SLE associated potentially functional variations from the three independent association signals in HLA-D region. (D) Haplotype association statistics in cases and controls with risk (red) and protective (blue) haplotypes highlighted. (E) Allelic bias in level of transcription for HLA-class II genes between SLE risk and non-risk alleles in 11 independent heterozygous donors (measured as shown in *Figure 6*). Number of RNA sequencing reads were compared between chromosome carrying risk (orange line) verses non-risk (blue line) allele for each class II gene. (F) MJ network analysis illustrating the relationships of risk and non-risk haplotypes based on 32 functional variations. SLE associated variants sitting exactly within specific protein binding motifs i.e. IRF4, CTCF and ZNF143 are highlighted with arrows.

the risk clade. Taken together, these results suggest that these DRB1 regulatory variations may not play a dominant role in the endophenotypes causing the association with SLE.

The SLE associations of regulatory variations spanning the entire HLA-D interval were assessed using composite haplotypes formed with 32 regulatory variants with strong SLE association signals that were derived from the three independent SLE associated signals. As shown in *Figure 8C*, these variants spanned a 116 Kb block containing DRB1, DQA1, and DQB1 and are all in tight LD. The composite analysis formed eleven haplotypes that accounted for more than 90% of all of the chromosomes identified within the EA panel (*Figure 8D*). As shown, HAP1 (p=2.38E-07, OR = 0.59) and HAP6 (p=1.01E-06, OR = 0.42) are protective, and HAP3 (p=4.56E-09, OR = 2.1), HAP2 (p=0.032, OR = 1.3), and HAP11 (p=0.0172, OR = 2.3) are risk. MJ analysis revealed a pattern similar to that obtained for XL9 haplotypes (*Figure 5F* versus *Figure 8F*). As shown in *Figure 8E*, an assessment of chromosome-specific transcription in MDM cultures from eleven heterozygotes for the rs9271593 (peak XL9 signal) revealed consistent and highly significant increases in transcription of HLA-DRB1, HLA-DQA1, and HLA-DQB1 for chromosomes carrying risk-associated variants. These results demonstrate that all of the regulatory haplotypes carrying the risk allele of rs9271593, which is the XL9 peak SNP (*Figure 8F*), transcribe higher levels of HLA-D class II genes than protective regulatory haplotypes.

## Comparing HLA class II alleles and HLA-D regulatory haplotypes in SLE susceptibility

The HLA-D sequence data allowed the imputation of standard HLA-DRB1, HLA-DQA1, and HLA-DQB1 class II alleles with four digit accuracy for the EA cohort, using algorithms and strategies described previously (*Morris et al., 2012*). The imputed HLA class II allele designations were used to assess the associations of HLA class II alleles with SLE in our EA cohort and to assess the relationships of these classical HLA-D class II alleles with the defined HLA-D regulatory haplotypes. The table in *Figure 9A* lists the SLE association statistics for the most strongly associated HLA class II alleles and HLA-D regulatory haplotypes in the EA cohort. As shown, HLA-DRB1_0301 and HLA-DQB1_0201 were the most strongly associated HLA-D class II alleles in this analysis, which is consistent with several previous studies of HLA-D associations with SLE (*Morris et al., 2012*; *Armstrong et al., 2014*; *Fu et al., 2011*; *Furukawa et al., 2014*; *Kim et al., 2014*; *Morris et al., 2014*; *Fernando et al., 2012*). The disease associations and odds ratios detected for HAP3 of the XL9 signal (*Figure 5F*) and HAP2 of the DQB1 promoter signal are equivalent with these HLA class II alleles. Similarly, as shown in the bottom of *Figure 9A*, both regulatory haplotypes and HLA class II alleles are strongly associated with protection from SLE. Thus, HAP4 of the DQB1 promoter signal, HAP6 of the XL9 signal, and HAP7 of the DRB1 signal all have potent association statistics and odds ratios for decreased frequencies in cases, as do the classic HLA-D class II alleles HLA-DQB1_0302, HLA-DRB1_0402, and HLA-DQA1_0301.

*Figure 9B* presents conditional analyses of the SLE associations contributed by the HLA-D regulatory haplotypes and the imputed HLA class II alleles. The top plot illustrates that XL9 regulatory polymorphisms can completely remove the associations of HLA-DRB1 and HLA-DQA1 imputation variants with SLE, but does not remove the DQB1 imputation or 5' region regulatory haplotype associations. Similarly, conditioning on the DQB1 regulatory haplotype removes the association of HLA-DQB1 imputation SNPs with disease, but has little effect on the imputation or regulatory variations in DRB1, DQA1, or XL9 (*Figure 9B*, middle panel). Finally, conditioning on HLA-DRB1 imputation

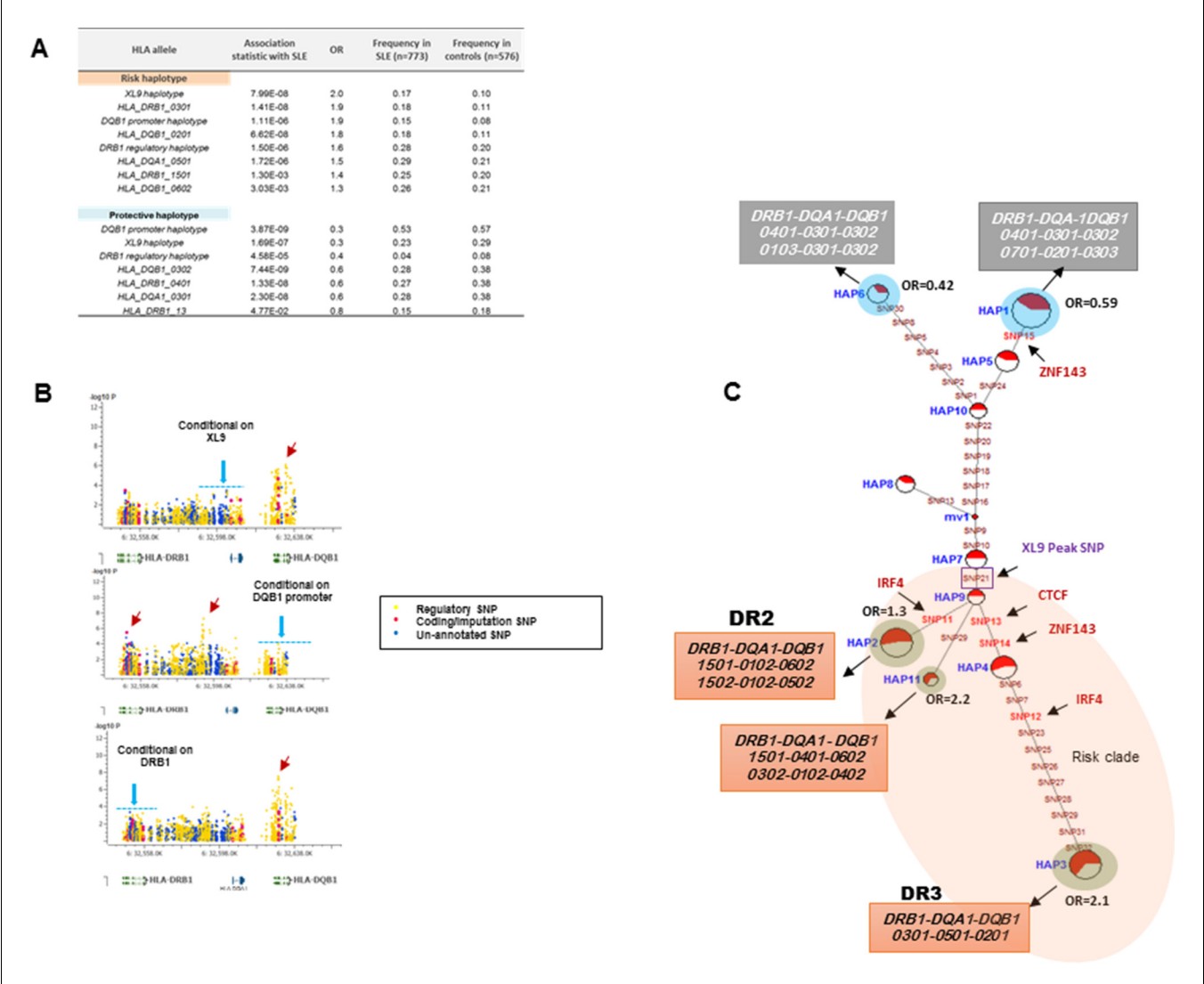

**Figure 9.** HLA-D regulatory haplotypes and classical HLA alleles. (**A**) SLE association statistics of regulatory and classical HLA alleles in this study. (**B**) Conditional analysis on peak regulatory signals in XL9, DQB1 and DRB1 regions. (**C**) Median-joining (MJ) network analysis of 32 regulatory variants spanning HLA-DRB1 to DQB1 region. SLE associated variants sitting directly on canonical binding motif of CTCF, IRF4 and ZNF143 transcription factor are indicated with arrows. The HLA DRB1-DQA1-DQB1 haplotypes associated with each of the risk and protective regulatory haplotypes are presented.

variants removes the association of XL9 regulatory variants and the HLA-DQA1 and HLA-DQB1 imputation SNPs, but does not significantly impact the association of the DQB1 regulatory haplotype with SLE. These analyses indicate that variations in XL9 and HLA-DRB1 class II alleles are in tight LD and represent a combined contribution to SLE, while variations in the segment 5' of DQB1 independently contribute to SLE susceptibility.

Finally, *Figure 9C* presents the MJ network formed by the composite HLA-D regulatory haplotypes (from *Figure 8F*) and overlays the imputed DRB1-DQA1-DQB1 HLA class II alleles present in individuals homozygous for the protective and risk HLA-D regulatory haplotypes. As shown, LD is very strong, but incomplete between the classical HLA class II alleles and the regulatory haplotypes. Notably, all homozygotes for regulatory HAP3 (peak risk) are also homozygous for DRB1_0301, DQA1_0501, DQB1_0201, which is the HLA-D class II haplotype found in the extended DR3 haplotype (*Kachru, 1984*; *Smolen et al., 1987*; *Hohler and Buschenfelde, 1994*; *Schur et al., 1990*; *Niu et al., 2015*). Similarly, regulatory risk HAP2 is predominantly associated with the DRB1_1501, DQA1_0401, DQB1_0602 haplotype, which has also been previously associated with susceptibility to

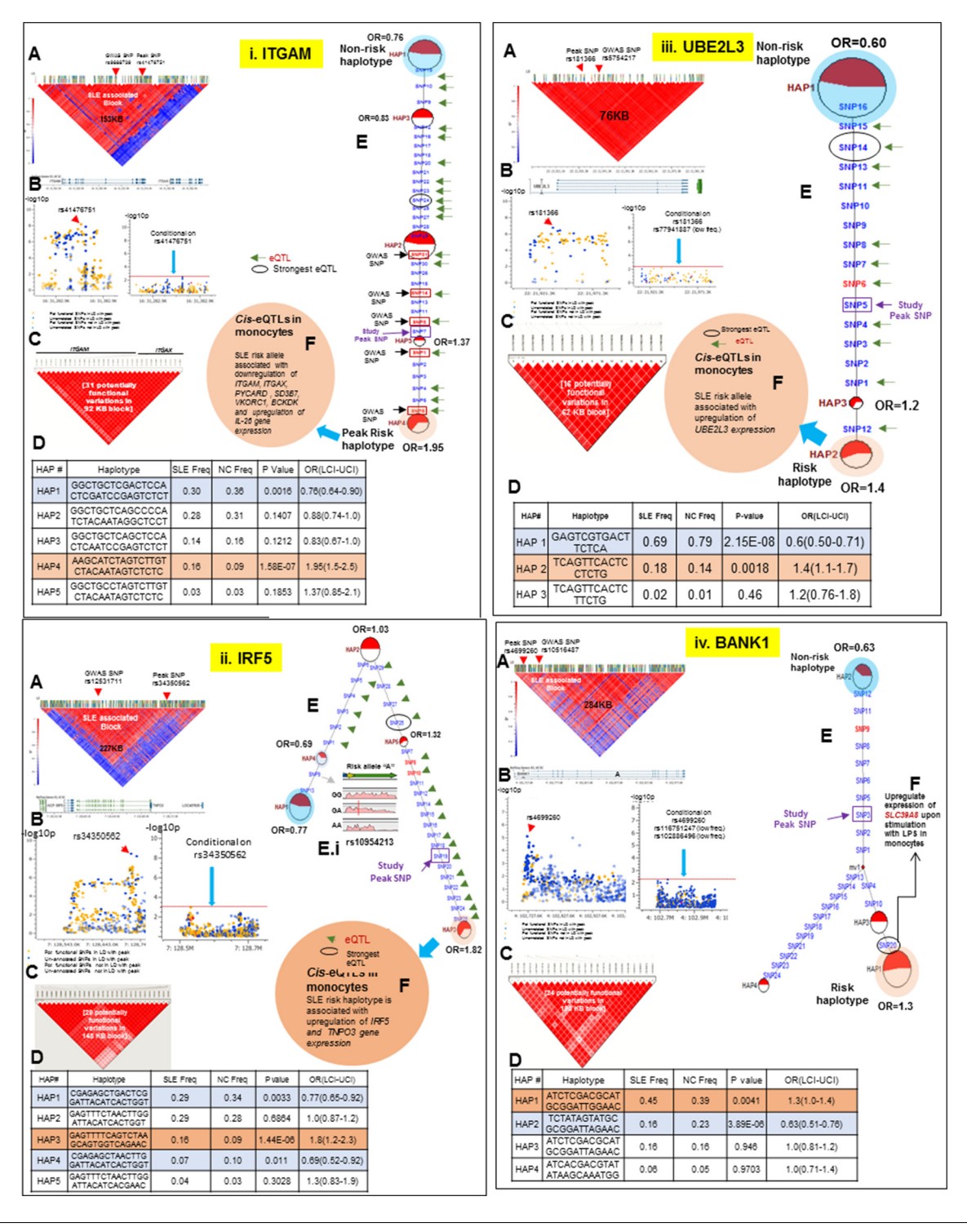

**Figure 10.** LD structure, haplotypes and MJ network analysis of *ITGAM, IRF5, UBE2L3 and BANK1.* Panel 10 (**i**) shows ITGAM, Panel 10 (**ii**) shows IRF5, Panel 10 (**iii**) shows UBE2L3 and Panel 10 (**iv**) shows BANK1 genetic association analysis. (**A**) LD structure of studied intervals generated with common (MAF≥10%) variants in 1349 samples, 221 in case of ITGAM, 400 in case of IRF5, 84 in case of UBE2L3 and 430 variants in case of BANK1. (**B**) Zoom Manhattan plot of all common variants in studied region showing SLE association levels and conditional analysis on peak SNP/s. (**C**) LD block based on potentially functional SLE associated SNPs which are used for downstream haplotype analysis. (**D**) Haploview generated haplotypes from functional variants. Frequencies in cases and controls and association statistics are provided. Risk (red) and protective (blue) haplotypes are color highlighted. (**E**) MJ networks analysis to illustrate divergence of risk and protective regulatory haplotypes. Haplotype with significant p value (p<0.05) are highlighted

*Figure 10 continued on next page*

*Figure 10 continued*

with red (risk) and blue (non-risk) color. Study peak SNP, previously known SLE GWAS tag SNP and eQTLs are indicated with arrows. (**F**) eQTL variations from public databases for variants in strongest risk haplotype.

SLE. Overall, the DR and DQ alleles that have been associated with SLE in previous studies of EA cohorts are found among the regulatory risk haplotypes and are absent from the protective clade (*Morris et al., 2012*; *2014*; *Niu et al., 2015*; *Ramos et al., 2010*). Taken together, these results are consistent with the strong LD within this small genomic segment of HLA and suggest that the regulatory variations and the peptide binding groove polymorphisms are two aspects of HLA-D diversification that are tightly intertwined within allelic lineages of the HLA-D region.

### Regulatory haplotypes in non-HLA risk loci

*Table 2* provides a summary for all sixteen SLE risk loci that have been analyzed in this study. Detailed analyses for the fourteen loci not discussed above are presented in *Figures 10–13*. Several characteristics of the genetic variations that underlie common SLE susceptibility alleles are revealed by these data. First, maximal risk for disease is associated with specific haplotypes typically composed of five or more functional variations that are in strong LD with the peak risk variant. The overwhelming majority of these variants are in regulatory elements (1199 of 1206, *Table 1*) and carry ENCODE scores indicating that they are potent functional polymorphisms. They occur as stable haplotypes within the EA population and are predicted to impact multiple endophenotypes. MJ analysis revealed that the risk and protective/non-risk haplotypes are typically at opposite ends of the networks (15 of 16 risk loci), indicating that significant variations in disease risk are most strongly associated with multiple functional changes. Furthermore, for some risk loci, multiple haplotypes are significantly associated with either risk or protection, but with varying odds ratios, indicating that a spectrum of functional haplotypes with varying disease risk contributions underlie the disease association of individual risk loci.

As tabulated in *Table 2*, peak risk haplotypes have greater odds ratios for SLE susceptibility than individual peak tagging SNPs from the original published GWAS studies and from the targeted association studies performed here. Overall, the peak risk haplotype had a higher odds ratio than the peak GWAS tagging SNP for 13 of 16 loci, resulting in an overall 17% increase in the average odds ratio for the sixteen loci tested. This result is consistent with theoretical predictions of the increase in odds ratio that would be achieved by specifically identifying causative variants in complex disease risk loci, thus supporting the presence of the causal variants of SLE within the identified functional haplotypes (*Gusev et al., 2013*; *Yang et al., 2010*).

Thirteen of the SLE risk loci characterized in detail here were identified previously by our group and others and the detailed sequence analyses of these risk loci has confirmed and extended these previous findings. Several of these loci contain long haplotypes, as discussed for *IRF5-TNPO3*, *ITGAM-ITGAX*, *TNFAIP3*, *UBE2L3* and *BANK1*, while *TNFSF4* and *XKR6* each contain two independent association signals in separate LD blocks (*Figures 10–13* and *Tables 1* and *2*). Our analyses found that regulatory haplotypes often contain variants impacting several eQTLs, chromatin structure and transcription regulatory elements. The ENCODE defined regulatory elements for *POLR2A*, *CTCF*, *IRF4*, *RELA*, *STAT5A*, *RFX5*, *RUNX3* were the most common regulatory elements affected by SLE associated variants (*Supplementary file 2*). Finally, three risk loci, *CCL22-CX3CL1* (*Figure 11. iii*), *ZGLP1-RAVER1* (*Figure 11.iv*), and *ICA1* (*Figure 12.i*), which were comparatively less well-studied for SLE association, were detected with strong statistical associations in this EA cohort (*Table 1*, *Supplementary file 2*). Detailed sequence analysis of these loci identified significant associations of these genes with SLE and identified SLE associated haplotypes impacting multiple regulatory components. More results on these loci have been incorporated into the relevant Figure legend for each risk locus.

## Discussion

These analyses provide a comprehensive assessment of the genomic variations associated with SLE disease alleles. We identified 345 regulatory variations impacting gene transcription within these loci

**Table 2.** Summary of SLE association and functional characteristics of peak variants and functional haplotypes for 16 SLE risk loci.

| Gene | Known GWAS (tag) SNP | GWAS reference | GWAS (tag) SNP OR | Study peak SNP | Study peak SNP OR | Peak risk-associated functional haplotype | Risk haplotype OR | Increase in OR of haplotype versus GWAS SNP | Increase in OR of haplotype versus study peak SNP | Related Figure in the manuscript | Strongest ENCODE effect cell line/tissue | eQTL data cell type/tissue |
|---|---|---|---|---|---|---|---|---|---|---|---|---|
| STAT4 | rs7574865 | Lee et al., 2012 | 1.4 | rs12612769 | 1.7 | ATTCCTT**GC** | 1.7 | 0.3 | 0 | Figure 4 | Mammary gland, Epithelial | Monocyte, macrophage |
| HLA-D | rs1150754 | Taylor et al., 2011 | 1.54 | rs9271593 (XL9) | 1.6910364 | CCCCTCCATC_ **TAGC**GGATGGCG AG**CATCGTCA** | 2.1 | 0.56 | 0.4089636 | Figure 8C-F | B-lymphocyte, lymphoblastoid | Monocyte |
| ITGAM-ITGAX | rs9888739 | Harley et al., 2008 | 1.6 | rs41476751 | 1.8696398 | AAGCATC TAGTCTT GTCTACAA TAGTCTCTC | 1.95 | 0.35 | 0.0803602 | Figure 10.i | B-lymphocyte, lymphoblastoid | Monocyte, Peripheral blood |
| IRF5-TNPO3 | rs12531711 | Chung et al., 2011 | 1.5 | rs34350562 | 1.7593583 | **GAGTT** TTCAGTCTA AGCAGT GGTCAGAA**C** | 1.8 | 0.3 | 0.0406417 | Figure 10.ii | Epithelial cell (Lung), B-lymphocyte | Monocyte, macrophage |
| UBE2L3 | rs5754217 | Chung et al., 2011 | 1.3 | rs181366 | 1.5217361 | TCA**GTT**CAC **TCCTCTG** | 1.4 | 0.10 | -0.1140361 | Figure 10.iii | Epithelial cell (Lung), B-lymphocyte | Monocyte |
| BANK1 | rs10516487 | Kozyrev et al., 2008 | 1.3 | rs4699260 | 1.25 | ATCTCGACGCA TGCGGA TT**GGAAC** | 1.3 | 0 | 0.05 | Figure 10.iv | Hela-S3, Epithelial, Fibroblast | Monocyte |
| TNIP1 | rs10036748 | Han et al., 2009; Galimberti et al., 2008 | 1.2 | rs62382335 | 1.37 | AATA**C**GGTC | 1.3 | 0.12 | -0.05 | Figure 11.i | B-lymphocyte, lymphoblastoid | Peripheral blood |
| TNFAIP3 | rs5029939 | Graham et al., 2008 | 2.2 | rs57087937 | 1.9092441 | GGGCAATCT TTGGGGCAAAT | 2.2 | 0.04 | 0.3307559 | Figure 11.ii | B-lymphocyte, lymphoblastoid, hepatocyte | no data |
| CCL22-CX3CL1 | rs223889 | Galimberti et al., 2008 | 1.4 | rs223889 | 1.45 | TATAAA**G**C | 1.5 | 0.05 | 0 | Figure 11.iii | B-lymphocyte, lymphoblastoid | Monocyte |
| ZGLIP-RAVER1 | rs35186095 | Present study | 1.3 | rs35186095 | 1.3173789 | TATAGTCT **GT**A**GG**ATG | 1.5 | 0.2 | 0.1826211 | Figure 11.iv | Fibroblast, K-562, HeLa-S3, B-lymphocyte | Monocyte |
| ICA1 | rs10156091 | Harley et al., 2008 | 1.3 | rs74787882 | 1.5 | G**GG**T | 1.5 | 0.2 | 0 | Figure 12.i | B-lymphocyte, lymphoblastoid | no data |
| BLK | rs1327113 | Hom et al., 2008 | 1.3 | rs7822109 | 1.26 | A**T**TT**GCCCCA** | 1.3 | 0 | 0.04 | Figure 12.ii | B-lymphocyte, lymphoblastoid | Monocyte, Peripheral blood |
| ETS1 | rs7932088 | Yang et al., 2010 | 1.2 | rs34516251 | 1.23 | GG**GC**GA | 1.4 | 0.2 | 0.17 | Figure 12.iii | B-lymphocyte, lymphoblastoid, Epithelial | Monocyte |

*Table 2 continued on next page*

*Table 2 continued*

| Gene | Known GWAS (tag) SNP | GWAS reference | GWAS (tag) SNP OR | Study peak SNP | Study peak SNP OR | Peak risk-associated functional haplotype | Risk haplotype OR | Increase in OR of haplotype versus GWAS SNP | Increase in OR of haplotype versus study peak SNP | Related Figure in the manuscript | Strongest ENCODE effect cell line/tissue | eQTL data cell type/tissue |
|---|---|---|---|---|---|---|---|---|---|---|---|---|
| TNFSF4 | rs2205960 | *Han et al., 2009* | 1.3 | rs4916313 | 1.3 | TCCATCTTCGA | 1.3 | 0 | 0 | *Figure 13.i* | Epithelial cell (Lung), Fibroblast, HeLa-S3 | no data |
| NMNAT2-SMG7 | rs2022013 | *Cunninghame Graham et al., 2011* | 1 | rs111487113 | 1.3 | TCACTAAC | 1.3 | 0.3 | 0 | *Figure 13.ii* | Primary Th1 T cells | no data |
| XKR6 | rs11783247 | *Harley et al., 2008* | 1.2 | rs7000132 | 1.2 | TGTCGCGGCTT | 1.2 | 0.03 | 0.03 | *Figure 13.iii* | Neuroblastoma, Mammary gland, Fibroblast | Monocyte |

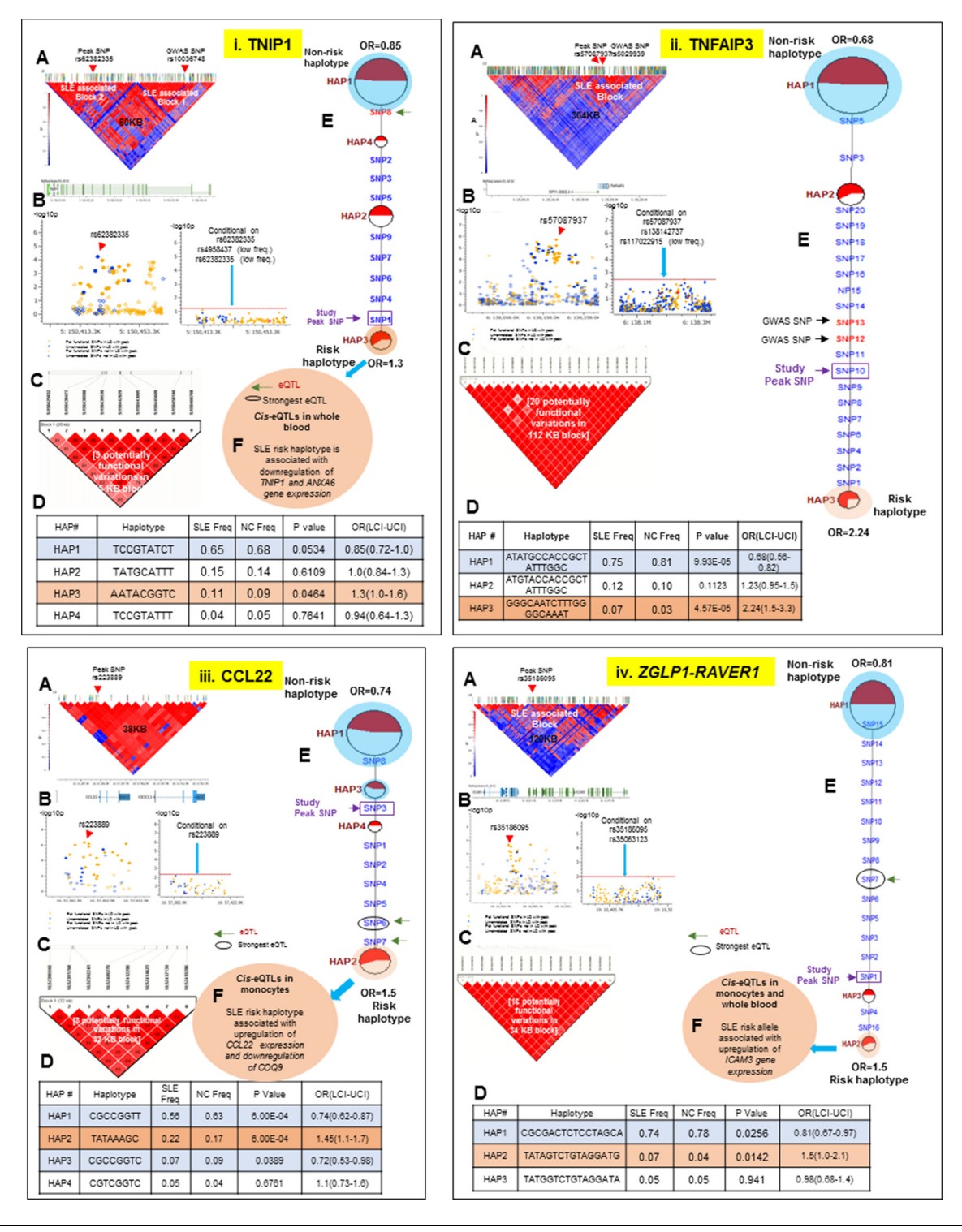

**Figure 11.** LD structure, haplotypes and MJ network analysis of *TNIP1, TNFAIP3, CCL22 and ZGLP1-RAVER1*. Panel 11 (**i**) shows TNIP1, Panel 11 (**ii**) shows TNFAIP3, Panel 11 (**iii**) shows CCL22 and Panel 11 (**iv**) shows ZGLP1-RAVER1 genetic association analysis. (**A**) LD structure of studied intervals generated with common (MAF≥10%) variants in 1349 samples, 140 in case of TNIP1, 356 in case of TNFAIP3, 30 in case of CCL22 and 126 variants in case of ZGLP1-RAVER1. (**B**) Zoom Manhattan plot of all common variants in studied region showing SLE association levels and conditional analysis on peak SNP/s. (**C**) LD block based on potentially functional SLE associated SNPs which are used for downstream haplotype analysis. (**D**) Haploview generated haplotypes from functional variants. Frequencies in cases and controls and association statistics are provided. Risk (red) and protective (blue) haplotypes are color highlighted. (**E**) MJ networks analysis to illustrate divergence of risk and protective regulatory haplotypes. Haplotype with

*Figure 11 continued on next page*

Figure 11 continued

significant p value (p<0.05) are highlighted with red (risk) and blue (non-risk) color. Study peak SNP, previously known SLE GWAS tag SNP and eQTLs are indicated with arrows. (F) eQTL variations from public databases for variants in strongest risk haplotype.

that exhibited stronger disease-associations than previously identified GWAS tagging SNPs (*Figure 3B*). Detailed analyses of the allelic architecture at these loci revealed that SLE disease alleles are haplotypes composed of multiple functional variations and that these variations often modulate several endophenotypes. This architecture is modeled in *Figure 14A*, which depicts the manner in which several functional variants in tight LD with GWAS tagging SNPs mediate transcriptional variations at multiple adjacent genes. As shown, the functional haplotypes identified in our analyses capture all of these causal variants within the LD block, which leads to the identification of a peak risk haplotype with increased disease association. Our analyses identified variants impacting multiple transcriptional changes at nine of the sixteen SLE risk loci, indicating that this level of complexity is prevalent among SLE risk loci. In this regard, we could only utilize our own MDM eQTL database and a few public eQTL databases for immune cell lineages in this analysis (datasets for monocytes, PBMC, and LBL are currently accessible, (*Fairfax et al., 2014*; *Raj et al., 2014*; *Westra et al., 2013*) and it is quite likely that these haplotypes will be found to have additional effects as more datasets become available.

*Figure 14B* presents the odds ratios for disease obtained with GWAS-defined tagging SNPs (*Harley et al., 2008*; *Graham et al., 2008*; *Adriantoet al., 2011*; *Taylor et al., 2011*) and peak risk haplotypes for the sixteen SLE loci analyzed. As shown, the odds ratio for disease obtained for the peak risk haplotype at each locus was consistently higher than that of the tagging SNP, leading to an average increase of 17% in odds of disease overall (*Table 2* and illustrated in *Figure 14B*). These results support the presence of most or all of the causal variations for disease susceptibility within the identified peak risk haplotypes for each locus. Interestingly, we identified both protective and risk haplotypes with nominal statistical significance (p<0.05) at fourteen of the sixteen risk loci analyzed, suggesting that both types of disease alleles are prevalent and contribute to population risk. MJ analyses consistently found peak risk and protective haplotypes at opposite ends of the network, which suggests that the combined effects of multiple regulatory variations may additively impact disease associations. Consistent with this, HLA-D, STAT4, IRF5, and CCL2-CX3Cl1 all have multiple haplotypes with different disease associations, suggesting that a spectrum of disease alleles with different impacts on susceptibility may occur at highly variable risk loci. However, some caution is appropriate when interpreting the significance of multiple intermediate risk haplotypes within a network, in that sample numbers for many haplotypes were often small. Consequently, a larger sample of SLE patients will be required for the detailed assessment of the disease risk attributable to all of the prevalent haplotypes at SLE risk loci.

These haplotypes represent stable polymorphisms within the EA population, with six or fewer haplotypes accounting for about 90% of the LD block regulatory sequences segregating in the EA population at individual risk loci (*Table 2* and *Figures 10–13*). We assessed this in more detail by determining the frequencies of the major risk haplotypes for HLA-D, STAT4, IRF5, ITGAM, and UBE2L3 in 2504 individuals derived from 26 global ethnic populations sequenced in the 1000 Genomes project (*Abecasis et al., 2010*; *2012*). As shown in *Figure 14C*, these five risk haplotypes are present at variable frequencies within all of the European, South American, and South Asian populations. However, they are much less frequent in the African populations sampled and, with the exception of UBE2L3, absent from the East Asian populations. This suggests that additional haplotypes will be identified during the analysis of specific ethnic groups as previously shown (*Kim-Howard et al., 2014*). In addition, although these five SLE risk haplotypes are predominantly found in European populations and populations with significant European admixture, they are also detectable with low frequencies in some African populations. Based on this distribution, it is likely that these haplotypes arose prior to human global colonization and that they were present with divergent frequencies in the ancestral founding populations of modern ethnic groups.

This dataset can also be used to assess the percentage of population disease risk that is attributable to the combined effects of all of these risk loci As tabulated in *Table 2*, the cumulative risk associated with the GWAS tagging SNPs for all sixteen loci sums to 6.04 fold, while the same value for

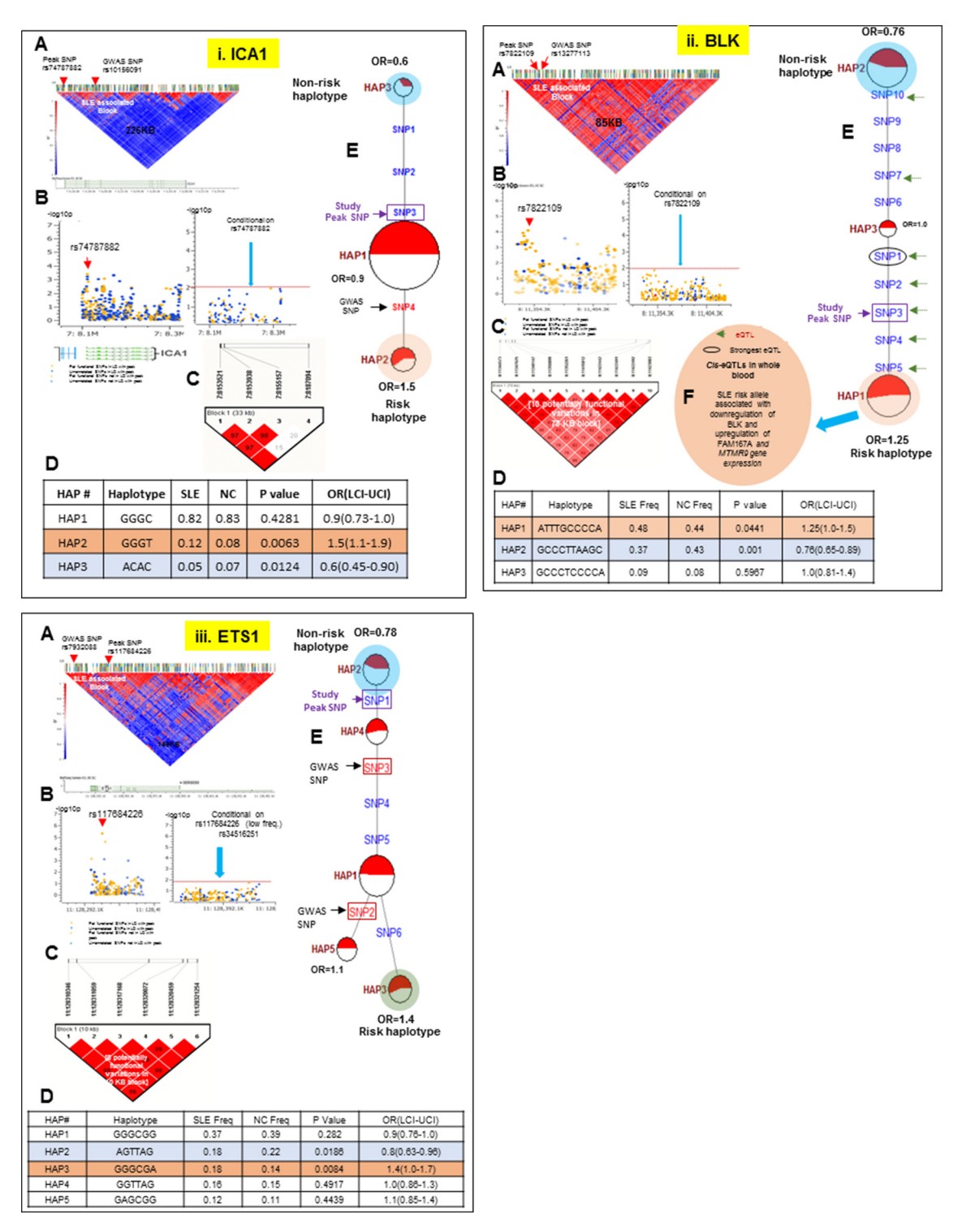

**Figure 12.** LD structure, haplotypes and MJ network analysis of *ICA1, BLK and ETS1*. Panel 12 (**i**) shows ICA1, Panel 12 (**ii**) shows BLK and Panel 12 (**iii**) shows ETS1 genetic association analysis. (**A**) LD structure of studied intervals generated with common (MAF≥10%) variants in 1349 samples, 370 in case of ICA1, 258 in case of BLK and 209 variants in case of ETS1 (**B**) Zoom Manhattan plot of all common variants in studied region showing SLE association levels and conditional analysis on peak SNP/s. (**C**) LD block based on potentially functional SLE associated SNPs which are used for downstream haplotype analysis. (**D**) Haploview generated haplotypes from functional variants. Frequencies in cases and controls and association statistics are provided. Risk (red) and protective (blue) haplotypes are color highlighted. (**E**) MJ networks analysis to illustrate divergence of risk and protective regulatory haplotypes. Haplotype with significant p value (p<0.05) are highlighted with red (risk) and blue (non-risk) color. Study peak SNP,

*Figure 12 continued on next page*

Figure 12 continued

previously known SLE GWAS tag SNP and eQTLs are indicated with arrows. (F) eQTL variations from public databases for variants in strongest risk haplotype.

all peak risk haplotypes is 8.8, indicating that improved resolution of disease alleles increases the disease risk and proportion of 'heritability' that is associated with these common disease alleles. Assuming that the contribution of all risk loci for SLE sum to 29 (*Alarcon-Segovia et al., 2005*; *Deapen et al., 1992*), then the sum of these sixteen loci would account for about one third of the genetic heritability for SLE. In this regard, a contentious debate concerning the contribution of 'common' (MAF > 0.05) versus 'rare' (MAF << 0.05) disease risk alleles to the overall heritability of common diseases has persisted among investigators in complex phenotype genetics for several years (*Raychaudhuri, 2011*; *Cirulli and Goldstein, 2010*; *Manolio et al., 2009*; *Pritchard and Cox, 2002*). Although it is clear that rare alleles contribute to disease susceptibility in small subsets of patients (*Hunt et al., 2013*; *Lee-Kirsch et al., 2007*; *Tang et al., 2014*; *Mitchell et al., 2002*), recent analytical studies have firmly established that common disease alleles are responsible for the bulk of the heritability for autoimmune diseases (*Yang et al., 2010*; *Visscher et al., 2010*; *Stahl et al., 2012*). It is likely that 'missing' heritability predominantly reflects an extensive genetic heterogeneity that underlies many common diseases.

An alternative method to measure the cumulative risk attributable to a specific collection of risk loci is via the calculation of population attributable risk (PAR) (*Zheng et al., 2008*; *Bruzzi et al., 1985*; *Rockhill et al., 1998a*; *1998b*; *Mezzetti et al., 1998*; *Natarajan et al., 2007*; *Claus et al., 1996*; *Kraft et al., 2009*; *Pepe et al., 2004*). This calculation utilizes the odds ratio for disease and the risk allele population frequency to calculate a weighted risk value for each locus and then combines them to assess their contribution to genetic risk for the population as a whole. As shown in *Figure 14B* and tabulated in *Supplementary file 3B*, the peak risk haplotypes at these sixteen risk loci account for 66% of the population attributable risk for SLE within this EA cohort (*Supplementary file 3A and 3B*). PAR and estimates of 'heritability' differ in that PAR calculations do not assume a specific level of population genetic risk (i.e. 29), but instead simply calculates the proportion of risk that <u>cannot</u> be accounted for by the variables assayed within the population studied, and thus determining the proportion of risk that is attributable to tested factors. The calculated PAR in this analysis indicates that these sixteen loci contribute a significant proportion of disease risk within our population. However, a larger cohort and broader list of risk loci will be essential to estimate genetic risk in all populations and account for a larger proportion of SLE heritability.

Our results provide as system using sequence analyses that can efficiently and accurately identify disease risk alleles within large populations. Further, we define a path forward for the development of useful genetic tools for assessing disease risk. The next phase of genetic analyses of autoimmune disease will involve assessing the functional properties of these disease alleles, sorting out their interactions during disease development, and developing analytical tools for the accurate quantitation of genetic risk for disease in individual genomes (*Ray and Hacohen, 2015*; *Ghodke-Puranik and Niewold, 2015*; *Lewi et al., 2015*; *Wang et al., 2015*; *Mohan and Putterman, 2015*).

## HLA-D polymorphisms, antigen presentation pathways, and autoimmune disease

The most intriguing result of our sequence analyses is the discovery of a strong association between SLE susceptibility and HLA-D polymorphisms that regulate HLA class II gene expression. The HLA-D region is consistently a potent susceptibility locus in autoimmunity and significant effort has focused on defining the molecular mechanisms that mediate autoimmunity in the context of specific HLA-D class II alleles (*Morris et al., 2012*; *2014*; *Armstrong et al., 2014*; *Kim et al., 2014*; *Niu et al., 2015*; *Graham et al., 2007*; *Cruz-Tapias et al., 2012*; *Todd et al., 1987*). Multiple genetic studies have identified coding variations in the peptide binding sites of MHC class II molecules as key genetic components of the disease associations, strongly supporting the hypothesis that allelic variations in the antigen presentation process underlie autoimmune disease (*Kim et al., 2014*; *Morris et al., 2014*; *Fernando et al., 2012*; *Raychaudhuri et al., 2012*). The dominant paradigm has been that the peptide binding regions of disease-associated HLA class II alleles have unique

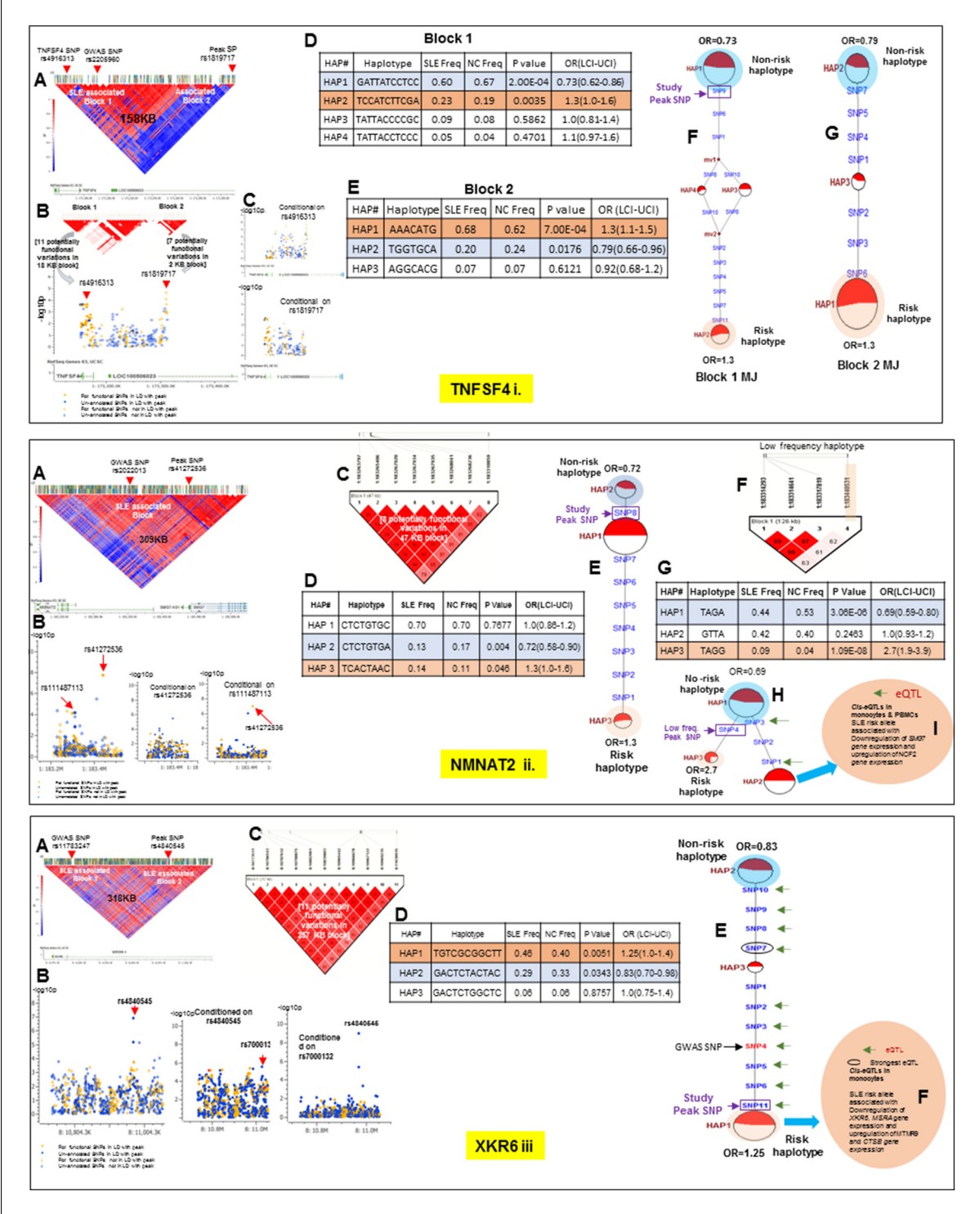

**Figure 13.** LD structure, haplotypes and MJ network analysis of *TNFSF4, NMNAT2 and XKR6*. Panel 13 (**i**) shows TNFSF4, Panel 13 (**ii**) shows NMNAT2 and Panel 13 (**iii**) shows XKR6 genetic association analysis. These three interval showed more than one independent LD block associated with SLE in our analysis. (**A**) LD structure of studied intervals generated with common (MAF≥10%) variants in 1349 samples, 152 in case of TNFSF4, 411 in case of NMNAT2 and 643 variants in case of XKR6. In case of TNFSF4, (**B**) shows two SLE associated LD blocks and zoom Manhattan plot of all common variants in studied region. (**C**) showing SLE association levels and conditional analysis on peak SNP/s. (**D and E**) Haploview generated haplotypes from functional variants in block 1 and block2, respectively. Frequencies in cases and controls and association statistics are provided. Risk (red) and protective (blue) haplotypes are color highlighted. Similarly, (**F and G**) shows MJ networks analysis to illustrate divergence of risk and protective regulatory haplotypes from block1 and block2, respectively. Haplotype with significant p value (p<0.05) are highlighted with red (risk) and blue (non-risk)

*Figure 13 continued on next page*

*Figure 13 continued*

color. Study peak SNP, previously known SLE GWAS tag SNP and eQTLs are indicated with arrows. In case of NMNAT2 (**13.ii**), (**B**) shows zoom Manhattan plot of all common variants in studied region showing SLE association levels and conditional analysis on peak SNP/s. Panel C: LD block based on potentially functional SLE associated SNPs which are used for downstream haplotype analysis. (**D**) Haploview generated haplotypes from functional variants. Frequencies in cases and controls and association statistics are provided. Risk (red) and protective (blue) haplotypes are color highlighted. (**E**) MJ networks analysis to illustrate divergence of risk and protective regulatory haplotypes. (**F**) LD block based on a low frequency SLE associated variant (**G**) Low frequency haplotype association analysis (**H**) MJ networks analysis with low frequency haplotype and (**I**) eQTL variations from public databases for variants in risk haplotype. In case of XKR6 (**13.iii**), (**B**) shows zoom Manhattan plot of all common variants in studied region showing SLE association levels and conditional analysis on peak SNP/s. (**C**) LD block based on potentially functional SLE associated SNPs which are used for downstream haplotype analysis. (**D**) Haploview generated haplotypes from functional variants. Frequencies in cases and controls and association statistics are provided. Risk (red) and protective (blue) haplotypes are color highlighted. (**E**) MJ networks analysis to illustrate divergence of risk and protective regulatory haplotypes. Haplotype with significant p value (p<0.05) are highlighted with red (risk) and blue (non-risk) color. Study peak SNP, previously known SLE GWAS tag SNP and eQTLs are indicated with arrows. (**F**) eQTL variations from public databases for variants in strongest risk haplotype.

peptide binding properties that present a novel spectrum of self-peptides or modified self-peptides in a manner capable of eliciting autoimmunity. Solid data supporting this mechanism have been developed by decades of experiments, notably for insulin peptides in autoimmune diabetes (*Unanue, 2014*) However, many studies have found that multiple self-antigens are recognized by T cells clones isolated from the earliest stages of disease development, suggesting that HLA-D associated autoimmunity is initiated against multiple self-antigens by a heterogeneous T cell response. Notably, SLE patients have a profound breach in immune tolerance and typically produce autoantibodies binding more than ten different self-antigens, with the diversity of autoantigens recognized increasing as individuals approach disease diagnosis (*Olsen and Karp, 2014*; *Arbuckle et al., 2003*; *Li et al., 2005*). Further, multiple HLA class II DR and DQ alleles are associated with SLE susceptibility, which indicates that HLA class II alleles with highly divergent peptide binding properties are capable of promoting disease development. In this regard, classic studies of the association of DR2 and DR3 with susceptibility to SLE have shown that DR2/DR3 heterozygotes are more strongly associated with disease susceptibility than the individual haplotypes (*Graham et al., 2007*). Taken together, these data suggest that SLE is associated with an extensive array of divergent HLA class II alleles that would be predicted to present a diverse array of self-peptides.

Our data indicate that all of the HLA-DR and -DQ alleles that are strongly associated with susceptibility to SLE are in strong LD with XL9 regulatory haplotypes that increase HLA class II gene transcription. Boss and co-workers have shown that XL9 contains CTCF elements that interact with cohesion molecules and other chromatin factors to assemble a transcriptional complex that brings multiple HLA class II promoters into close proximity with an array of transcription factor binding sites (*Majumder et al., 2006*; *2008*). The XL9 model in *Figure 15A.i*, which is adapted from a model presented by *Majumder et al. (2008)*, illustrates the key role of chromatin configuration in the coordinated transcription of HLA class II genes. Consistent with the chromatin structure effects of XL9, our data indicate that transcriptional variations are chromosome specific in HLA-D heterozygotes, with polymorphisms in the XL9 regulatory haplotype modulating transcription of DR and DQ genes in a cis-specific fashion. This indicates that the level of transcription dictated by XL9 will be specific for the adjacent HLA-DR or DQ allele, thus making expression levels an additional allele-specific facet of HLA-D class II molecules. Classic studies have demonstrated allele-specific variations in the expression levels of MHC class II molecules in murine MHC heterozygotes and shown that these levels strongly impacted the stimulation of antigen-specific T cells in autoimmune disease models (*Ridgway et al., 1998*). Our data suggest that these early studies revealed an important facet of MHC diversity that strongly impacts the development of autoimmunity.

XL9 contains the peak association signal with SLE in our sequencing dataset, indicating that these regulatory haplotypes play an important role in the development of SLE autoimmunity. We suspect that the failure to detect XL9 associations in previous GWAS reflects the low frequency of many of the associated variants and the paucity of SNPs in this region on the commonly utilized SNP typing arrays (see *Figure 5.iii & iv*). Our data demonstrate that individuals homozygous for XL9 HAP3 (risk) variations have more than two-fold higher surface expression for HLA-DR and DQ molecules at baseline which increases to 4-fold after stimulation with TLR-ligand on monocyte-derived dendritic cells

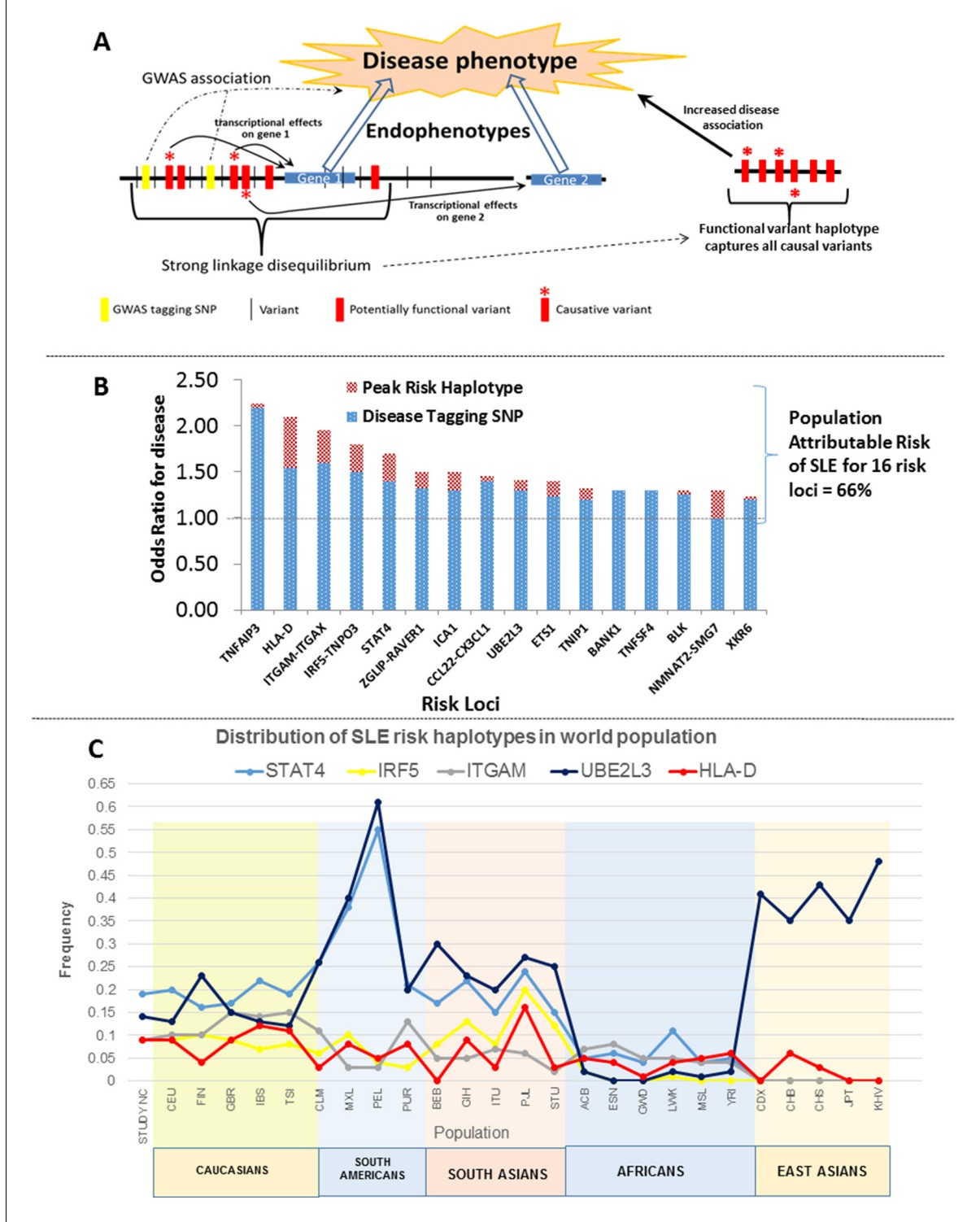

**Figure 14.** Model of allelic architecture for functional variations in common disease risk loci. (A) A working model of the architecture of the variations within common disease risk loci. Disease associated tagging SNPs associate an LD block with a disease phenotype. Within this LD block, multiple variations are in tight LD, including nonfunctional, functional, and causal variants. Causal variants potentiate the disease phenotype by modulating endophenotypes. In this model, causal variants impact two adjacent genes, one of which is not located within the LD block, both of which contribute endophenotypes towards disease. Haplotype and MJ analysis using functional variants in tight LD with original tagging SNP define haplotypes that contain all of the causal variants. The peak risk haplotype defines a disease allele with increased disease association in comparison to the original GWAS tagging SNP. (B) A plot of all of the odds ratios attributable to the GWAS tagging SNP (blue bars) versus the peak risk haplotype (additional red

*Figure 14 continued on next page*

*Figure 14 continued*

bar) for each of the sixteen risk loci analyzed in detail. A consistent gain in odds ratio for SLE was obtained with regulatory haplotypes that averaged 17% in the present study. (C) Frequency of *STAT4, IRF5-TNPO3, ITGAM-ITGAX, UBE2L3 and HLA-D* SLE risk haplotypes among our own study and 26 ethnic populations characterized in the 1000 Genomes project. The x-axis of the graph shows population groups and y axis show frequency of haplotypes.

than HAP1 (protective) homozygotes (*Figure 7*). This increase is maintained during the maturation of these dendritic cells via TLR7/8 stimulation, thus supporting the functional significance of these transcriptional variations to immune mechanisms known to impact immune response activation. Potent polymorphisms in the binding motif of the IRF4 transcription factor are in the final MJ branch to the SLE-associated XL9 HAP2 and HAP3 haplotypes and it is likely that these two polymorphisms are causal and contribute significantly to this expression change. Recent studies by Vander Lugt et al (*Vander Lugt et al., 2014*) have demonstrated that IRF4 is a key component of the transcriptional regulation of HLA class II molecules in dendritic cells and that upregulation of MHC class II molecules strongly promotes susceptibility to autoimmunity in an animal model. Based on these data, we hypothesize that the XL9-mediated increase in surface expression of HLA-DR and DQ in dendritic cells is predominantly responsible for the association of XL9 regulatory haplotypes with susceptibility to SLE.

The extensive diversification of HLA-D regulatory elements has implications well beyond the association of these polymorphisms with SLE. HLA class II molecules are expressed on a variety of immune cell lineages, including monocytes, macrophages, dendritic cells, B cells, activated T cells and thymic epithelial cells. Expression levels are tightly controlled by a variety of transcription factors unique to these cell lineages and their expression impacts a variety of functional processes in the immune system (*Cresswell, 1994*; *Krawczyk et al., 2004*; *Steimle et al., 1994*; *Reith et al., 2005*). For example, increased surface expression of class II molecules is an essential event in the maturation of dendritic cells, in that higher surface expression is crucial to the increased capacity of mature dendritic cells to effectively present antigens to naïve T cells (*Cella et al., 1997a*; *1997b*; *Banchereau and Steinman, 1998*; *Pierre, 1997*). As tabulated in *Figure 15A.ii*, twenty-three transcription factor binding sites in XL9 and fourteen in the DQB1 5' segment are strongly modified by the extensive variations present within these genomic segments. Overall, we identified a total of 1651 functional variants in XL9 and 1912 functional variants in the DQB1 5' segment, indicating that the regulatory characteristics of these transcriptional complexes will be highly diversified among HLA-D haplotypes. This level of polymorphism is readily comparable to that observed in the codons for the peptide binding regions of the HLA class II molecules. Interestingly, as shown in *Figure 15B*, both the protective and risk haplotypes that we identified for the entire HLA-D region are present with varying frequencies throughout the global population. These findings are consistent with the characteristics of other highly polymorphic regions of HLA, including the HLA-D class II coding alleles, and indicate that these regulatory haplotypes have evolved together with the HLA class II coding regions over long periods. Given all of these characteristics, we propose that these regulatory variations represent an essential and highly selected characteristic of the diversification of HLA-D that strongly impacts a variety of immune functions.

These regulatory HLA-D haplotypes probably have divergent effects on the expression levels of HLA class II molecules in different cell lineages. That is, XL9 HAP3 clearly upregulates HLA class II expression in the myeloid lineage, however, that may not be true for HAP3 B cells, T cells, or thymic epithelial cells. In addition, XL9 variants impact the function of a variety of transcription factors whose activity is modified by innate system activation signals in specific cell lineages, indicating that modulations of HLA class II expression levels by activation signals may also be differentially associated with individual haplotypes. Taken together, these features indicate that the regulatory polymorphisms identified in this study will have multifaceted effects on the adaptive immune response and probably result in a significant diversification in the functioning of HLA class II antigen presentation among HLA haplotypes.

Finally, a variety of eQTL studies have identified the HLA complex as a 'master' regulatory complex that impacts the expression of genes throughout the genome (*Fehrmann et al., 2011*; *2012*). An analysis of the available eQTL databases identified a total of 72 genes whose transcription

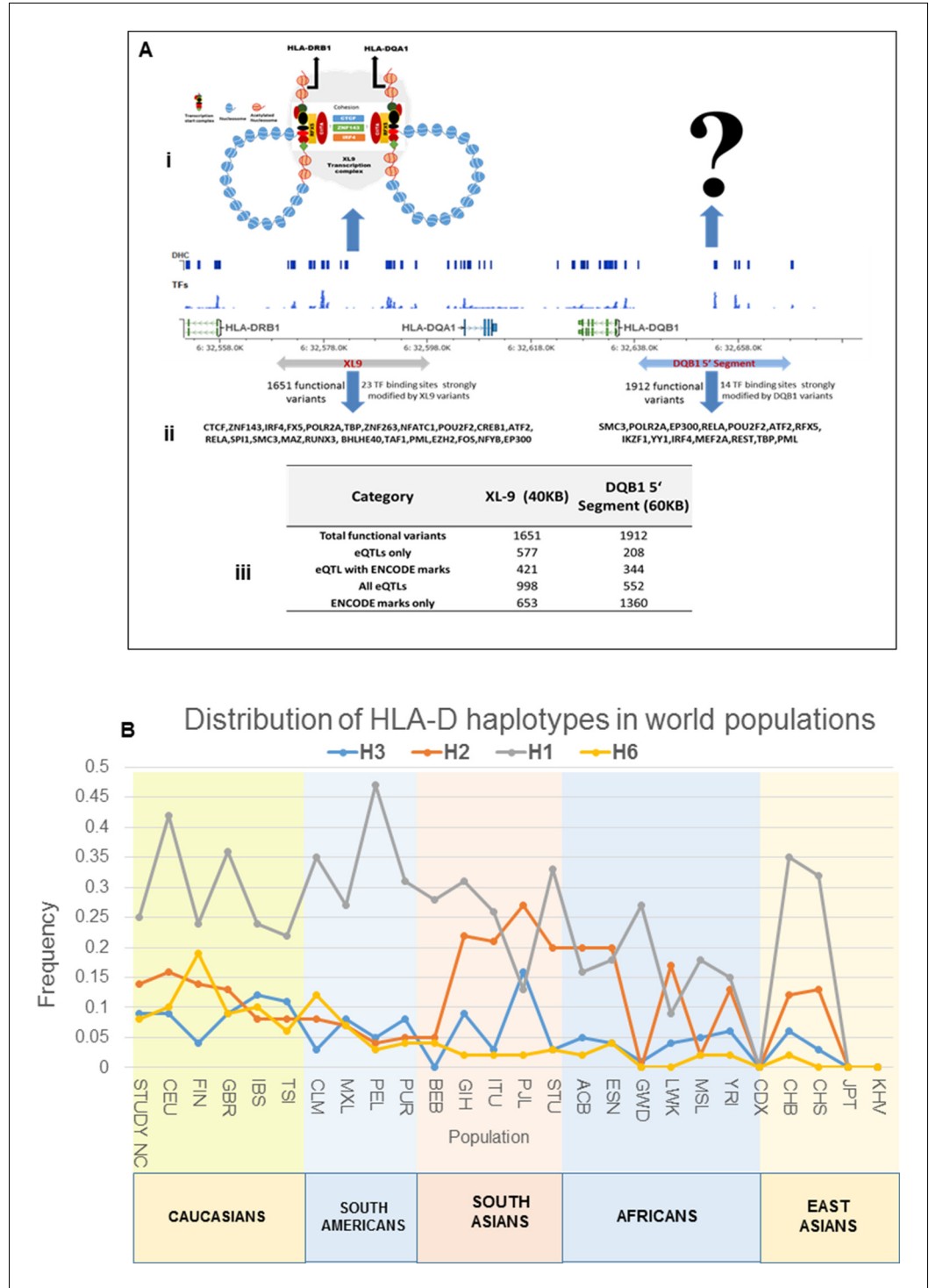

**Figure 15.** Model of chromatin architecture and transcription regulatory elements in XL9 and DQB1 segments. (**A**) (i-ii) A model showing the XL9 transcription complex and three important proteins (CTCF, IRF4 and ZNF143) which may be impacted by SLE associated genetic variants hitting canonical motifs in XL9 region (Adapted from *Majumdar et al., 2008*). The chromatin structure of the regulatory complex produced in the DQB1 5' segment is hypothetical and currently unknown. A chromosomal map of HLA-DRB1 through HLA-DQB1 region showing ENCODE defined regulatory marks, eQTLs and most strongly impacted transcription factors by XL9 and the DQB1 5'segment is shown below these models. The transcription factor binding sites impacted by functional variations within these regions are shown below the molecular map. (**A**) (iii) A table listing the numbers of and characteristics

*Figure 15 continued on next page*

Figure 15 continued
of functional variants in these two regulatory regions of HLA-D. (**B**): Global distribution of the major risk and protective haplotypes from the composite HLA-D region analysis.

levels are reported to be modulated by variations in the HLA-D region. Many of these eQTL targets are encoded in other segments of the HLA complex, as well as 35 that are located on other chromosomes. A variety of immune system genes are included in this list and *Figure 16* illustrates the pattern of up and down expression that distinguishes the HAP3 risk haplotype from the HAP1 and HAP6 protective haplotypes. As shown, essentially all of the HLA class II molecules and a variety of gene products involved in antigen processing, peptide loading, and surface expression are up-regulated by the HAP3 risk haplotype. This result illustrates the extensive functional variations that are associated with a single HLA-D risk haplotype within the EA population. Whether these eQTLs reflect the formation of remarkably intricate transcriptional complexes, or (more likely) very strong LD throughout the HLA complex remains to be determined. However, these results indicate that regulatory polymorphisms in HLA-D affect a plethora of immune system genes that are involved in various pathways of the adaptive immune system. It is likely that these regulatory variations are an integral element of the functional diversification of HLA and that they will ultimately be found to modulate functions throughout the immune system.

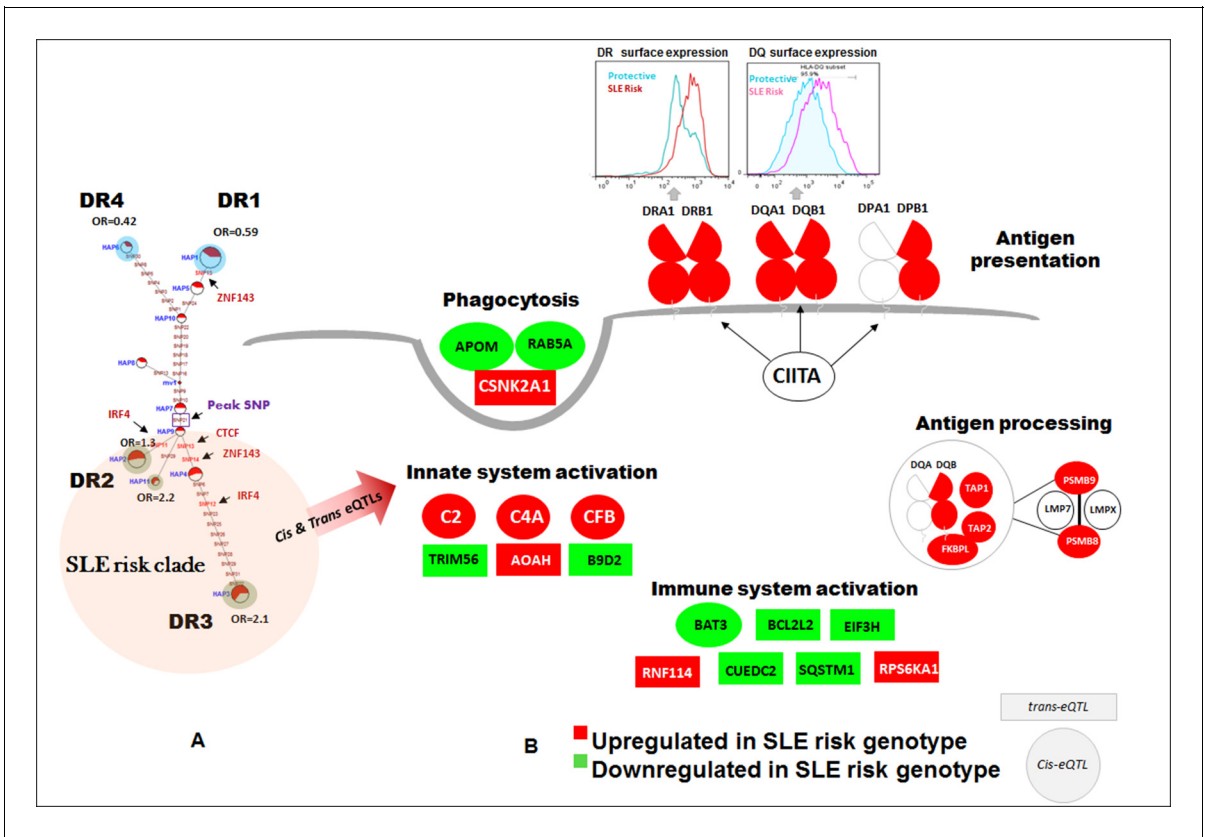

**Figure 16.** SLE risk haplotype upregulates the antigen presentation pathway (APP). All of the composite HLA-D haplotypes within the risk clade (highlighted in red) contain eQTL variants reported to impact 72 genes in the publicly available eQTL datasets utilized in this study. The patterns of increased or decreased transcription associated with all of these haplotypes is modeled on the left, with red indicating increased expression and green indicating decreased expression relative to the protective haplotypes shaded in blue. All of the HLA-DR, HLA-DQ, and HLA-DP class II molecules, along with a variety of gene products involved in the APP pathway are upregulated in all SLE risk haplotypes. A variety of other genes in the immune system, including some with known associations with SLE susceptibility (C2, C4A) are also modulated.

## Materials and methods

### Targeted sequencing of SLE risk loci

Genomic sequencing libraries were prepared from 1775 SLE patient or control samples contributed by 5 collaborating sites in the U.S.A and Europe (*Supplementary file 1A*). All subjects gave their written informed consent and research protocols and methods employed were approved by the UT Southwestern Institutional Review Board. More than 50% of SLE cases and all of the control samples used in the present study were new recruitments and have not been used in any previous association or GWAS on SLE. Target enrichment and deep sequencing was carried out in the UT Southwestern Medical Center IIMT Genomics Core. 1 ug picogreen measured genomic DNA was sonicated using *Covaris* S220 platform to generate 300–400 bp genomic fragments. The sequencing libraries were generated using TruSeq (Illumina) or KAPA Biosystem library preparation kits (KK8232). Each sample was ligated with custom designed Illumina-compatible adaptors with unique 6 base barcodes following the kit manufacturer's protocol. The custom target enrichment array (Illumina Inc. San Diego, CA, www.illumina.com) was designed to capture the complete genome sequence of 28 confirmed or potential SLE risk loci (*Supplementary file 1B*). The Illumina custom enrichment system theoretically captured sequence information for ~99.94% of the ~4.4 Mbs of genome targeted in these risk loci. The enriched libraries were sequenced using a paired-end 100 bp protocol to produce 1–2 Gb of high quality data per sample.

### Sequence alignment and variant calling

Sequence reads were demultiplexed and each sample's reads were aligned to the human genome (HG19) using BWA-MEM, with base quality recalibration and local realignment performed with the Genome Analysis Toolkit (GATKv2) (*Li et al., 2009*; *McKenna et al., 2010*; *DePristo et al., 2011*). As illustrated in *Figure 1A*, target enrichment was highly specific and efficient, typically resulting in >70% of reads on target and resulting in >128X average coverage for the 28 risk loci analyzed (*Supplementary file 1B*). *Figure 1B* illustrates that coverage within the targeted segments was comprehensive, with relatively uniform read depth throughout the non-repetitive regions. In general, continuous sequences could be derived for >85% of the targeted intervals in assembled sequences, with only extended regions (i.e. >1 Kb) of highly repetitive sequences failing to assemble.

Variant analysis with the GATK Haplotype caller identified a total of 215880 variations in the targeted regions using analytic technologies and thresholds analogous to those of the 1000 Genomes Project (*Supplementary file 1B*) (*Abecasis et al., 2010*; *2012*).

### Defining high quality variants in the EA population

As listed in *Figure 1G,H*, we used additional criteria to filter this dataset to create an ethnically-matched, case-control cohort with uniform coverage and high quality variant calls. *Figure 1C* illustrates the read depth and balanced allele representation of variants that passed all filters and *Figure 2A* presents principal component analysis of sample ethnicity in comparison to standardized populations from the HAPMAP dataset (*International HapMap et al., 2003*). We started with 1775 samples, all of which were sequenced with targeted array. Of these, 88 samples had missing case/control status information and 249 were PCA outliers as they did not cluster with HapMap CEU reference population in principal component analysis. Of the remaining 1438 PCA pass samples, 11 and 5 were excluded due poor call rate (<85%) and being duplicate, respectively. Furthermore, 73 samples were excluded due to poor sequencing fold coverage (<25x, n=54) and significant p value (p>0.001) of HWE in controls (n=19). Thus, 1349 samples which included 773 SLEs and 576 normal controls passed all quality criteria and were used for genetic association analysis. Application of these filters defined 1349 samples of EA ancestry and identified 124,552 high quality variants, of which 114,487 are single nucleotide variations (SNVs) and 10,065 are insertion/deletion (In/Del) polymorphisms. Data supporting the accuracy of these variant calls is provided in *Figure 1F*, which presents Sanger sequence validation of several key variants. In addition, the concordance of sequence-based SNV calls versus SNV calls with the Illumina Immunochip was >99.8% for samples with >25X average coverage (*Figure 1G*), which supports the overall accuracy of variant detection throughout the genomic segments assayed. This sequence-based variant database, which identifies an average of one variant every 39 basepairs through 4.4 Mb of genomic DNA in 28 validated SLE

risk *loci*, provides a comprehensive assessment of genomic diversity at SLE risk loci in the EA population. Raw sequencing data (FASTQ files) for all targeted intervals in 1349 individuals is available on request (www.utsouthwestern.edu/labs/wakeland/about/contact.html).

## Variant annotation

Variants were annotated using multiple databases cataloguing the functional properties of human genomic variation, including the recent phase 3 release from the 1000 genome study, the PolyPhen/ SIFT coding region database, the most recent release of the ENCODE database, and several recent eQTL databases for immune cell lineages. The outcome of these analyses for all variants in the final dataset is summarized in *Figure 2B,C*. Our sequence analyses identified 70070 previously annotated variations and 54482 unannotated or novel variations. All but 76 of the novel variations were low frequency or rare (MAF<0.05) within the EA cohort, which is consistent with expectations (*Supplementary file 1C*). Functional annotation determined that about 40% of the variants in the dataset were potentially functional, most of which were categorized as regulatory based on their localization into ENCODE-defined regulatory segments or inclusion in eQTL datasets (*Figure 2B.i-iv*). *Figures 2C.i-iii* shows summary statistics on just common coding, regulatory and novel variants.

## Immunochip genotyping and Sanger sequencing

A subset (n=536) of study samples was also genotyped with the Immunochipv1, an Illumina infinium genotyping chip which contains 196524 genomewide markers. SNP concordance analysis was done between sequencing and immunochip genotypes in order to validate the quality of sequencing calls. Raw image files from immunochip array were imported into Genome Studio (GS version 1.9.4) and SNPs were called. The genotype outputs from Genome Studio were then imported into SNP & Variation suite (SVS version 7.6.8 win64) for further quality control (QC) and downstream association analysis. In addition, about 35 SLE associated SNPs from various targeted genes were also confirmed by Sanger sequencing method.

## Quality control criteria

Quality control filters were applied to both samples and markers. All the duplicates and those with call rate <85% were excluded from the analysis. Principal component analysis (PCA) was done to remove any population stratification. About 2902 markers (MAF≥0.05) present in both Omni1 HapMap data set and our sequencing data were used for PCA (*Figure 2A*). PCA clusters was further confirmed by doing another PCA based on ~26000 markers present on Omni1 HapMap data set and Immunochip v1.0 on subset samples (n=408). Samples that did not cluster with HapMap CEU reference population were excluded after visual inspection of plots. SNP concordance was done between sequencing and immunochip data on subset of samples (n=408), which showed a genotype concordance rate ≥98% at ≥25x fold coverage. Therefore, samples with fold coverage <25x were excluded from downstream analysis. For genetic association tests, autosomal markers showing HWE p<0.001 in controls were excluded.

## Genetic association tests and haplotype analysis

Of the 23,805 common (MAF≥0.05) markers detected in 28 targeted loci, 15,582 quality pass variants were used for genetic association tests. A basic allelic association test was performed with 1349 PCA confirmed and age matched European cases (n=773) and controls (n=576). The association test was controlled for genomic inflation using Golden Helix scripts where we first determined uncorrected genomic inflation factor $\lambda$ value which was 3.0. We further corrected data for batch effects and stratification with PCA using numeric association and regression analysis in Golden Helix. Finally, we corrected association results using this inflation value. This removed observed genomic inflation in association results. Chi-square p values were further corrected for gender bias, using the covariate regression module in SVS, Golden Helix software. A total of 5561 markers across 28 loci showed significant (p<0.05) association with SLE. The LD structure and haplotype analysis was performed in SVS, Golden Helix and using Haploview v4.2. Regulatory haplotypes were generated on potentially functional variants with strong LD (≥0.8) to the study peak and/or previously known SLE tagging SNPs. An allele was defined as potentially functional if it is a coding variation i.e. non-

synonymous, synonymous, UTR or splice variant, and or an ENCODE's histone mark, transcription factor binding site, DNase I hypersensitivity clusters or an expressed quantitative trait loci (eQTL).

## Population attributable risk (PAR) calculation

PAR has been used in genome-wide association studies (*Ziegler and studies, 2009*; *Bonnelykke et al., 2013*; *Wang et al., 2010*). It combines information on risk allele frequencies and genotypic relative risks to estimate the excess fraction of cases that would not occur if no one in the population carried the risk allele. The case/control design of present study provided authors an opportunity to apply odds ratio (OR) and PAR to calculate the relative risk of SLE disease. As it has already been shown and present study also confirms that SLE is a polygenic disease, assessment of cumulative disease risk by calculating joint PAR from all the loci is a reasonable approach. First we performed conditional analysis on peak SNP from all the 16 loci analyzed in the present study. We observed residual SLE association after conditioning on each locus (*Supplementary file 3A*), which suggests significant and cumulative contribution of each locus to SLE risk within this population. Then, we used 16 SLE risk haplotypes to assess the percentage of population disease risk that is attributable to the combined effects of all of these risk loci (*Supplementary file 3B*). PAR was calculated using methods applied in other complex diseases (*Zheng et al., 2008*).

## Production of monocyte-derived macrophages and dendritic cells

Human peripheral blood mononuclear cells (PBMCs) were enriched by density gradient centrifugation of peripheral blood from healthy human donors through a Ficoll-Hypaque gradient. Monocytes were isolated from PBMC either by negative selection using an EasySep Human Monocyte Enrichment Kit (STEMCELL Technologies), or by cell culture dish adherence as plastic adherence method. For monocyte isolation by adherence, PBMCs were plated in tissue culture treated dishes and incubated for 2 hours at 37°C in a humidified $CO_2$ incubator. Non-adherent cells were discarded by washing three times. For the generation of monocyte-derived dendritic cells (MDDCs) or human monocyte-derived macrophages (MDMs), monocytes were cultured in RPMI-1640 with 10% FBS, 2 mM L-glutamine, 10 mM HEPES, 1 mM sodium pyruvate, 100 U/ml penicilin, 100 μg/mL streptomycin supplemented with 100 ng/ml GM-CSF and 50 ng/ml IL-4 or 50 ng/ml M-CSF, respectively. The culture media which contained fresh GM-CSF and IL-4 or M-CSF were replaced every 2 days. MDDCs and MDMs were harvested on day 7. MDDCs or MDMs were seeded in 6 or 12 well plates at a density of $1 \times 10^6$ or $5 \times 10^5$ cells/ well, respectively, treated and incubated with 10 ug/ml R848 for 18 hr. R848 (InvivoGen, tlrl-r848) is an imidazoquinoline compound with potent anti-viral activity. This low synthetic molecule activates immune cells via the TLR7/TLR8 MyD88-dependent signaling pathway (*Hemmi et al., 2002*)

## Flow cytometry

MDDCs were incubated with anti-HLA-DR FITC (clone G46-6) for 20 min on ice and washed three times with PBS. MDDCs were acquired on a FACS Calibur (BD Biosciences) and data analyzed using FlowJo software. For HLA-DQ staining, MDDC's were stimulated with 1 ug/ml R848 for 4, 8 and 18 hr. Cells were harvested and washed twice with PBS. Staining was done with HLA-DQ-FITC (Clone: Tu169) in PBS for 30 min on ice followed by two washes with PBS and ran on BD FACSCALIBUR. Data was analyzed using FlowJO software.

## RNA-Seq data production and analysis

RNA was extracted using TRIZOL (Life Technologies) and RNeasy Mini Kit (QIAGEN) according to the manufacturer's protocol. RNA quantity and purity was assessed on a NanoDrop 2000 spectrophotometer (Thermo Fisher Scientific), and integrity was measured on an Agilent Bioanalyzer 2100 (Agilent Technologies). RNA-seq libraries were prepared with the Illumina TruSeq RNA Sample Preparation kit (Illumina) according to the manufacturer's protocol. Libraries were validated on an Agilent Bioanalyzer 2100. Six RNAseq libraries were sequenced on a SE50 (single end 50 base pair) Hiseq 2500 lane, which yielded an average of about $30 \times 10^6$ reads/sample. We used CLC Genomics Workbench 7 for bioinformatics and statistical analysis of the sequencing data. This approach used by CLC Genomics Workbench is based on method developed by Mortazavi et al. (*Mortazavi et al., 2008*). Human Genome GRCh37 was used as reference sequence. The reference has 33615 genes

and 30842 transcripts. All uniquely mapping reads to the genes were counted. Alignment with mismatch cost of '2', Insertion cost '3' Deletion cost of '3' was used. The maximum number of hits for a read was set to 1 meaning that only reads those maps uniquely were considered. The steady state expression of various genes was calculated in terms of RPKM values. For eQTL analysis RPKM values were normalized as described previously (*Dozmorov and Lefkovits, 2009*; *Dozmorov et al., 2011*) as well as for population stratification or batch effect and cis-eQTL results were corrected for gender and ethnicity.

## Public databases used

We have accessed multiple public databases to validate and functionally annotate sequencing variants identified in the present study. We used DNA and RNA sequencing data-based variants from the 1000 Genome Project samples (http://www.1000genomes.org/); and downloaded DNA sequencing data from the phase III dataset (ftp://ftp.1000genomes.ebi.ac.uk/vol1/ftp/release/20130502/) for haplotype analysis of 2504 genomic samples from the global human population. Similarly, FASTQ files of RNA sequencing data (http://www.geuvadis.org/web/geuvadis/RNAseq-project) of lymphoblastoid cell lines derived from 369 Europeans were downloaded and used for HLA class II expression analysis. Our analysis of this data is shown in *Figure 7D* as a heatmap. The ENCODE database (www.encodeproject.org) was used to annotate variants for transcription factors binding motif, DNase hypersensitivity cluster and histone marks. Similarly, the RegulomeDB database (http://regulomedb.org/) was used to annotate potentially regulatory variations. Finally, UCSC genome browser (https://genome.ucsc.edu/) was used to generate custom tracks on sequencing variants and for the general visualization of study data.

## Source data text
### ITGAM-ITGAX

We replicated association with previously reported GWAS SNPs in *ITGAM* [Integrin, Alpha M (Complement Component 3 Receptor 3 Subunit)] and *ITGAX* [Integrin, Alpha X (Complement Component 3 Receptor 4 Subunit)] gene and identified new common variants showing strong association with SLE in our analysis (*Supplementary file 2*). The peak associated SNP rs41476751 [OR (LCI-UCI) = 1.87 (1.5–2.2), p=7.84E-09] was mapped to the same LD block where previous GWAS SNP was tagged (*Figure 10.i*). Conditioning on peak SNP removed all the disease association (*Figure 10.i B*). We found 30 potentially functional variations including four known SLE GWAS SNPs in strong LD with peak (*Supplementary file 2*). Haplotypes were then derived on these 31 potentially functional variations and genetic association analysis was done in cases vs. controls (*Figure 10.i D*). Association results showed that haplotype 4 (HAP4) which include all previously reported GWAS SLE alleles also carry multiple potentially functional variations (eQTLs) and pose strongest SLE risk [OR (LCI-UCI) = 1.95 (1.5–2.5), p=1.58E-07]. Median-joining network analysis illustrated accumulation of multiple potentially functional alleles in the risk haplotype (*Figure 10.iE*). Analysis of published eQTLs data showed that risk haplotype carries multiple regulatory variations which cumulatively contribute to the down-regulation of *ITGAM*, *ITGAX* and *PYCARD* genes (*Figure 10.iF*). Interestingly, we observed that OR of strongest haplotype (HAP4) was significantly increased as compared to known GWAS tag (rs9888739) as well as study peak SNP (*Table 2*).

### IRF5-TNPO3

We replicated previous SLE associations at interferon regulatory factor 5 (*IRF5*) and transportin 3 (*TNPO3*) gene region and also identified many new potentially functional variants showing strong association with disease (*Supplementary file 2*). Multiple association signals at this locus were mapped to *IRF5, TNPO3* and *TPI1P2* gene region with peak SNP rs34350562 [OR (LCI-UCI) = 1.76 (1.4–2.1), p=2.86E-09] mapped near *TNPO3* gene (*Figure 10.iiA*). *Figure 11.iiB-C* shows Manhattan plot of SLE association and conditional analysis on peak SNP, respectively. Total of 29 SLE associated potentially functional variants were identified in the associated LD block which also included four previously known GWAS tags (10.iiC). Haplotypes were generated based on all 29 variants and haplotype association analysis was done. These results showed that haplotype 3 (HAP3) which carries previously reported GWAS SLE SNPs and multiple potential functional alleles identified in present sequencing study pose the greatest risk for SLE [OR (LCI-UCI) = 1.8 (1.2–2.3), p=1.44E-06]

(*Figure 10.iiD*). Median-joining network analysis showed that risk haplotype HAP3 differs from the non-risk haplotype (HAP1) by multiple functional variations which regulates transcription of local genes (*Figure 10.iiE*). Further analysis of eQTL SNPs showed that SLE risk alleles were associated with upregulation of *IRF5* and *TNPO3* expression in monocytes, PBMCs and MDMs (*Figure 10.iiF*). In addition, a 3'UTR truncation of *IRF5* gene rs10954213 (SNP9) was also a part of strongest risk haplotype. Our analysis shows that there are multiple potential functional variants present at this locus which contributes to the SLE susceptibility.

## UBE2L3

*UBE2L3* (Ubiquitin-conjugating enzyme E2L3) gene is a known SLE susceptibility gene. We observed strong association signal at this locus and identified multiple regulatory variants (*Supplementary file 2*). The peak association was observed with SNP rs181366 [OR (LCI-UCI) = 1.5 (2.0–3.1), p=1.98E-07] in same LD block where previously reported GWAS tag was located (*Figure 10.iiiA*). We replicated previous SLE associations at this locus and identified many new associations. In addition to *UBE2L3*, strong association signal were also mapped to *SCUBE1* (Signal Peptide, CUB Domain, EGF-Like 1) gene with SNP rs4647815 [OR (LCI-UCI) = 1.7 (1.2–2.3), p=5.46E-06] and *YDJC* (YdjC Homolog (Bacterial) gene with SNP rs2298429 [OR (LCI-UCI) = 1.5 (1.2–1.8), p=9.26E-07]. Manhattan plot of association and conditioning analysis are shown in *Figure 10.iiiB*. There was low frequency variant which was not in strong LD with peak common variant. We identified 15 potentially functional variations in strong LD with peak SNP including previously known GWAS tags for SLE. Haplotypes were derived based on these 16 markers and haplotype association analysis was done (*Figure 10.iiiC*). As shown in *Figure 10.iiiD*, haplotype 2 (HAP2) which carries GWAS SLE allele and multiple potentially functional alleles was the greatest risk haplotype [OR (LCI-UCI) = 1.41 (1.1–1.7), p=1.80E-03]. Median-joining network analysis illustrated that risk haplotype (HAP2) differs from non-risk haplotype (HAP1) by 16 putatively functional changes (*Figure 10.iiiE*). Further, SNPs with published eQTL effects shows that SLE associated risk haplotype associate with upregulation of UBE2L3 expression in peripheral blood data (*Figure 10.iiiF*). Comparison of tag SLE SNP verses haplotype suggest that present regulatory risk haplotype accounts for increased disease risk (OR=1.4) than known GWAS SNP rs5754217 (OR=1.3) alone (*Table 2*).

## BANK1

We observed peak association with an intronic SNP rs4699260 [OR (LCI-UCI) = 1.5(1.3–1.8), p=8.54E-06] in *BANK1* (B-cell scaffold protein with ankyrin repeats 1) gene. Previously reported SLE associations were replicated and some new variants were identified in same LD block (*Figure 10.ivA*). Association signal was in the range of suggestive genomewide significance (*Figure 10.ivB*). We also observed few low frequency variants in the same block. *Figure 10.ivB* also shows conditional analysis based on peak common variants and two low frequency variants. Annotation of SLE associated variants for possible potentially functional effects revealed 23 variants in strong LD with peak associated SNP as well as previously known GWAS tags (*Figure 10.ivC*). So, we performed haplotype analysis on all 24 potentially functional variations (*Supplementary file 2*). Haplotype association analysis showed that haplotype 1 (HAP1) is the strongest risk haplotype [OR (LCI-UCI) = 1.3 (1.0–1.4), p=0.004] (*Figure 10.ivD*), which differs from non-risk haplotype by at least 13 potentially functional variations (*Figure 10.ivE*). An interesting eQTL (SNP 20, rs17208914) was observed in risk haplotype which associate with the upregulation of *SLC39A8* gene expression upon stimulation by LPS in monocyte.

## TNIP1

Multiple strong association signals were observed with *TNIP1* (TNFAIP3-interacting protein 1) gene (*Supplementary file 2*). The peak associated SNP in our analysis was rs62382335 [OR (LCI-UCI) = 1.37 (1.0–1.8), p=6.42E-05] located in a different block than previously known GWAS tag (*Figure 11.iA*). Still, both were in strong LD with each other due to long LD block. We observed a modest association signal at this locus (*Figure 11.iB*). Conditional analysis showed that in addition to peak common variant, there were few low frequency variants associated with SLE (*Figure 11.iB*). We generated haplotypes on 9 potentially functional variations including peak and GWAS tag SNP (*Figure 11.iC*). The haplotype analysis showed that haplotype 3 (HAP3) is the strongest risk haplotype

[OR (LCI–UCI) = 1.30 (1.0–1.6), p=0.04] (*Figure 11.iD*). Median-joining network analysis showed that risk haplotype (HAP3) differs from non-risk haplotype (HAP1) by 9 potentially functional changes (*Figure 11.iE*), which include an eQTL associated with downregulation of *TNIP1* and ANXA6 (Annexin A6) gene expression in PBMCs (*Figure 11.iF*).

## TNFAIP3

Multiple strong association signals were observed in *TNFAIP3* (tumor necrosis factor, alpha-induced protein 3) gene in present study. The peak associated SNP rs57087937 [OR (LCI–UCI) = 1.9 (1.4–2.5), p=2.05E-06] is a ENCODE defined regulatory variant with strong potential to impact binding of several transcription factors in intron 1 region of *TNFAIP3* gene. We also replicated previously reported association with rs5029939 [OR (LCI–UCI) = 2.0 (1.5–2.9), p=2.86E-05] which was located in the same LD block (*Figure 11.iiA*). In addition, some new potentially regulatory variants were observed in our analysis (*Supplementary file 2*). *Figure 11.iiB* shows Manhattan plot of SLE association and conditional analysis, respectively. We identified 20 potentially functional variations including peak SNP and known GWAS alleles (SNP rs5029930 (OR=1.5, p=7.74355E-05), rs7750604 (OR=1.5, p=8.37827E-05), rs719149 (OR=1.5, p=0.0001) and rs719150 (OR=1.5, p=0.0001) in strong LD (D'≥0.8) (*Figure 11.iiC*). Next, haplotypes were derived on these 20 markers and association analysis was done. Haplotype association results showed that haplotype 3 (HAP3) which also carries previously known SLE alleles confer strongest risk [OR (LCI–UCI) = 2.2 (1.4–2.9), p=4.57E-05] to SLE (*Figure 11.iiD*). Median -joining network analysis shows that risk haplotype (HAP3) differs from non-risk haplotype (HAP1) by 20 potentially functional variations (*Figure 11.iiE*).

## CCL22-CX3CL1

Strong association signals were observed in *CCL22* [Chemokine (C-C Motif) Ligand 22(CCL22)] and *CX3CL1* [Chemokine (C-X3-C Motif) Ligand 1] genes (*Supplementary file 2*).The peak association was observed with SNP rs223889 [OR (LCI–UCI) = 1.5(1.2–1.7), p=4.93E-07] in CCL22 gene (*Figure 11.iiiA*). SLE association statistics is shown in Manhattan plot (*Figure 11.iiiB*). All the SLE association was gone after conditioning on peak SNP. Peak SNP is an eQTL associated with downregulation of COQ9 gene expression in peripheral blood (*Westra et al., 2013*). Seven potentially functional variations were identified in strong LD with peak signal (*Figure 11.iiiC*). Four common haplotypes were identified based on 8 potentially functional variations. The haplotype association test results showed that HAP2 [OR (LCI–UCI) = 1.5 (1.1–1.7), p=6.00E-04] confer strongest SLE risk as compared to HAP1 which is protective [OR (LCI–UCI) = 0.74 (0.62–0.87), p=6.00E-04] (*Figure 11.iiiD*). Median -joining network analysis showed that risk haplotype (HAP2) differs from non-risk haplotype (HAP1) by 8 potentially functional variations (*Figure 11.iiiE*). eQTL analysis showed that strongest risk haplotype is associated with upregulation of *CCL22* and downregulation of *COQ9* gene expression in monocytes (*Figure 11.iiiF*).

## ZGLP1-RAVER1

We observed multiple SLE association signals at this locus which were mapped to various local genes including ZGLP1 (zinc finger, GATA-like protein 1), FDX1L (Ferredoxin 1-like), RAVER1 (Homo sapiens ribonucleoprotein, PTB-binding 1), ICAM1 (Intercellular adhesion molecule 1), and TYK2 (Tyrosine kinase 2) genes (*Supplementary file 2*). Peak SNP rs35186095 [OR (LCI–UCI) = 1.32 (1.0–1.6), p=2.10E-04] was sitting in the middle of a big LD block (*Figure 11.ivA*), mapped to intron 9 of *ZGLP1* gene. Manhattan plot illustrate the SLE association statistics as shown in *Figure 11.ivB*. The conditioning on first and second peak variant removed all the observed association. We identified 16 potentially functional variations in SLE associated block and derived haplotypes on these (*Figure 11.ivC*). Haplotype association analysis showed that haplotype 2 (HAP2) is the strongest risk haplotype [OR (LCI–UCI) = 1.5 (1.0–2.1), p=0.01] (*Figure 11.ivD*). Median-joining network illustrates number of variants that differs between risk (HAP2) and non-risk (HAP1) haplotypes (*Figure 11.ivE*). The eQTL data from literature suggests that SLE risk haplotype is associated with upregulated expression of ICAM3 in human monocytes and PBMCs (*Figure 11.ivF*).

## ICA1

We observed modest association of *ICA1* (Islet Cell Autoantigen 1, 69kDa) gene with SLE in our study. The peak association was observed with SNP rs74787882 [OR (LCI-UCI) = 0.67 (0.49–0.92), p=1.61E-03] which was mapped to the previously associated LD block (*Figure 12.iA*). We replicated previous association with SNP rs10156091 [OR (LCI-UCI) = 1.5 (1.1–1.9), p=0.02]. Manhattan plot shown in *Figure 12.iB* shows the strength of SLE association at this locus. All the association was gone after conditioning on peak variant. Haplotypes were derived based on four potentially functional variants in the SLE associated LD block (*Figure 12.iC*). Haplotype association test results showed that haplotype 2 (HAP2) is the strongest risk haplotype in *ICA1* (*Figure 12.iD*). Median-Joining network shows that HAP2 is differed from non-risk haplotype HAP3 by four potentially functional variants (*Figure 12.iE*). Associated data is provided in *Supplementary file 2*.

## BLK

The SLE association signal at BLK (B lymphoid tyrosine kinase) gene was moderate in our analysis. However, we did replicate previously reported associations at this locus. SLE associated peak SNP rs7822109 [OR (LCI-UCI) = 0.79 (0.68–0.92), p=8.57E-05] is a potentially functional (ENCODE, eQTL effects) variant. It is in strong LD (D'$\geq$0.8) with previously reported SLE GWAS SNPs (*Figure 12.iiA*). Manhattan plot in *Figure 12.iiB* shows SLE association strength in present study and conditional analysis. We derived haplotypes based on 10 potentially functional variations in this LD block and performed haplotype association test (*Figure 12.iiC*). Results shows that haplotype 1 [OR (LCI-UCI) = 1.25 (1.0–1.5), p=0.04] pose strongest risk for SLE (*Figure 12.iiD*). Several eQTLs cumulated into SLE risk haplotype 1 associate with down regulation of *BLK* and upregulation of *FAM167A* and *MTMR9* genes in peripheral blood (*Figure 12.iiE-F*). Associated data is provided in *Supplementary file 2*.

## ETS1

Peak SLE association at *ETS1* (V-ETS avian erythroblastosis virus E26 oncogene homolog 1) gene was observed with a low frequency SNP rs117684226 [OR (LCI-UCI) = 0.47 (0.35–0.63), p=4.87E-06] located in the previously implicated LD block (*Figure 12.iiiA*). In addition, rs34516251 was the strongest common SLE risk alleles located in the same block. Manhattan plot in *Figure 12.iiiB* shows the SLE association statistics. Conditional analysis revealed that peak low frequency variant and strongest common allele accounts for all the observed association at this locus in the present study (*Figure 12.iiiB*). We identified 6 potentially functional variants in SLE associated LD block and derived haplotypes on them (*Figure 12.iiiC*). Haplotype analysis showed that haplotype 3 (HAP3) is the greatest risk haplotype [OR (LCI-UCI) = 1.4 (1.0–1.7), p=0.008] (*Figure 12.iiiD*), which differs from non-risk haplotype HAP2 by 5 variants as illustrated in Median-Joining network (*Figure 12.iiiE*). Associated data is provided in *Supplementary file 2*.

## TNFSF4

We observed two independent association signal in *TNFSF4* (Tumor necrosis factor (ligand) superfamily member 4) gene region (*Figure 13.iA*). First signal was SNP rs4916313 [1.34 [OR (LCI-UCI) = 1.34 (1.1–1.5), p=0.0002] in previously associated LD block (block 1). This SNP is a strong regulatory variant defined by ENCODE data. In block 1, we identified 11 potentially functional variations in strong LD with peak SNP. Manhattan plot shows the strength of SLE association in block1 and block2 (*Figure 13.iB*). Conditioning analysis on peak variants in each block showed that both the signals are independent of each other (*Figure 13.iC*) Haplotypes association analysis in block1 showed that haplotype 2 (HAP2) was the greatest risk haplotype [OR (LCI-UCI) = 1.33 (1.1–1.6), p=0.003] (*Figure 13.iD*), which differs from HAP1 by 11 variations (*Figure 13.iF*). In block 2, where SNP rs1819717 was the strongest association [OR (LCI-UCI) = 1.36 (1.1–1.6), p=2.29E-05] was mapped to an uncharacterized gene LOC100506023. Haplotype analysis based on seven potentially functional SNPs showed that HAP1 which carries multiple potentially functional variants pose strongest risk for SLE from this region (*Figure 13.iE*). Median-joining network analysis shows distribution of various risk alleles between risk and non-risk haplotypes from block 2 (*Figure 13.iG*).

Associated data is provided in *Supplementary file 2*.

## NMNAT2

We observed a modest association of polymorphisms in *NMNAT2* (Nicotinamide nucleotide adenylyl transferase 2) gene with SLE in present study (*Supplementary file 2*). The peak associated SNP was though a low frequency variant rs41272536 [OR (LCI-UCI) = 2.9 (2.0–4.0), p=1.87E-08] but reached genome wide level of significance. It was pretty much in LD with previously reported tag SNP (*Figure 13.iiA*). This SNP is a ENCODE defined strong regulatory variant which can impact binding of several transcription factors, enhancers and insulators. Another strongly associated common variant was SNP rs111487113. Conditional analysis on peak and strongest common alleles suggested that two effects are independently associated with SLE (*Figure 13.iiB*). We identified 7 potentially functional variations in strong LD with the strongest common variant and derived haplotypes (*Figure 13.iiC*). Haplotype association analysis showed that haplotype 3 (HAP3) was the strongest risk haplotype [OR (LCI-UCI) = 1.3 (1.0–1.6), p=0.04] (*Figure 13.iiD-E*). We also derived haplotypes based on three potential functional variants in modest LD with low frequency peak variant and found that HAP3 which carry low frequency risk allele have strong SLE risk (OR=2.7) for SLE (*Figure 13.iiF-H*). Also, we found that eQTL variations in low frequency risk haplotype were associated with down regulation of SMG7 (Homo sapiens smg-7 homolog, nonsense mediated mRNA decay factor) and upregulation of NCF2 (Neutrophil cytosolic factor 2) gene expression in monocytes and PBMCs (*Figure 13.iiI*).

## XKR6

We observed more than one association signals at XKR6 (XK, Kell blood group complex subunit related family, member 6) gene (*Supplementary file 2*). The peak associated SNP was rs4840545 [OR (LCI-UCI) = 1.95 (1.5–2.5), p=1.27E-07] was a low frequency marker mapped to a different LD block than previously reported GWAS tag (*Figure 13.iiiA*). SNP rs7000132 was the strongest common allele associated with SLE at this locus but it was not in complete LD with peak low frequency variant as shown by the conditional analysis (*Figure 13.iiiB*). We identified 10 potentially functional variations in strong LD with rs7000132 including previously reported SLE associated rs11783247 SNP and derived haplotypes (*Figure 13.iiiC*) Haplotype association test showed that HAP1 was the greatest risk haplotype associated with SLE [OR (LCI-UCI) = 1.25 (1.0–1.4), p=0.005] (*Figure 13.iiiD*). Median-Joining network analysis illustrated that risk haplotype 1 differs from non-risk haplotype 2 by several eQTL SNPs (*Figure 13.iiiE*), which are associated with down regulation of XKR6, MSRA gene expression and upregulation of MTMR9 and CTSB gene expression in monocytes (*Figure 13.iiiF*).

### Statistical analyses

We used SNP & Variation suite (SVS) of Golden Helix (version 7.6.8 win64, *Golden Helix, Inc., Bozeman, MT, www.goldenhelix.com) for genetic analysis*. SNP conditioning analysis was done using regression module of SVS. Haploview v.2 software was used for visualization of LD plots and haplotype analysis (Barrett et al., 2005). GraphPad Prism 6.0 software was used for statistical analysis and graphics. Correlations between continuous variables were determined using Pearson's r in GraphPad Prism 6.0. Discontinuous variables were compared by Fisher's exact test. P values <0.05 were considered significant.

## Acknowledgements

We thank the many SLE patients and control participants whose sample contributions were essential for these studies. We also thank all of the personnel in the IIMT Genomics Core at UT Southwestern Medical Center for their excellent technical support and participation. These studies were supported by multiple grants from the NIH, the Alliance for Lupus Research, and the Walter M. and Helen D. Bader Center for Research on Arthritis and Autoimmune Diseases.

## Additional information

### Funding

| Funder | Grant reference number | Author |
|---|---|---|
| Alliance for Lupus Research | | Prithvi Raj<br>Ran Song<br>Shaheen Khan |
| Walter M. and Helen D. Bader Center | | Edward K Wakeland |
| NIH Office of the Director | AR055503 | David R Karp<br>Quan Zhen Li<br>Patrick M Gaffney<br>Edward K Wakeland |
| NIH Office of the Director | AI045196 | Edward K Wakeland |
| NIH Office of the Director | AR058959 | Graham B Wiley<br>Jennifer A Kelly |

The funding from above listed agencies supported sample collection, data generation, data analysis, etc., and man power for the present study

### Author contributions

PR, EKW, Conception and design, Acquisition of data, Analysis and interpretation of data, Drafting or revising the article, Contributed unpublished essential data or reagents; ER, MM, GBW, JAK, CC, COJ, Acquisition of data, Analysis and interpretation of data, Contributed unpublished essential data or reagents; RS, SK, KV, BZ, ID, Acquisition of data, Analysis and interpretation of data, Drafting or revising the article; BEW, Analysis and interpretation of data, Drafting or revising the article, Contributed unpublished essential data or reagents; CA, CL, FC-J, BRL, NJO, CKG, CAW, JBH, SKN, JAJ, BPT, QZL, Conception and design, Acquisition of data, Analysis and interpretation of data; CP, Conception and design, Analysis and interpretation of data, Drafting or revising the article; DRK, PMG, Conception and design, Acquisition of data, Analysis and interpretation of data, Drafting or revising the article

### Author ORCIDs

Edward K Wakeland, http://orcid.org/0000-0002-7107-0992

### Ethics

Human subjects: All the study subjects gave their written informed consent for the study. All the research protocols and methods employed were approved by UT Southwestern Institutional Review Board.

## Additional files

### Supplementary files

• Supplementary file 1. (A) SLE patients and controls analyzed in this study (B) Genomic intervals of SLE risk loci targeted for sequencing (C) Characteristics of unannotated/novel common variants (MAF$\geq$0.05) detected in this study (D) Peak association signal detected for each of the 28 SLE risk loci (E) Association status of previously published GWAS tagging SNPs (F) Sequencing variants that are strongly associated with SLE.

• Supplementary file 2. Summary of functional properties of all variants in tight LD with disease tagging SNPs used for haplotype analysis.

• Supplementary file 3. (A) Conditional analysis on SLE associated 16 peak SNPs.(B) Calculation of joint *PAR* on 16 SLE risk loci.

## Major datasets

The following previously published datasets were used:

| Author(s) | Year | Dataset title | Dataset URL | Database, license, and accessibility information |
|---|---|---|---|---|
| Fairfax BP, Humburg P, Makino S, Naranbhai V, Wong D, Lau E, Jostins L, Plant K, Andrews R, McGee C, Knight JC | 2014 | Innate Immune Activity Conditions the Effect of Regulatory Variants upon Monocyte Gene Expression | http://www.ncbi.nlm.nih. gov/pmc/articles/ PMC4064786/ | Publicly available at the NCBI Gene Expression Omnibus (Accession no: PMC4064786). |
| Raj T, Rothamel K, Mostafavi S, Ye C, Lee MN, Replogle JM, Feng T, Lee M, Asinovski N, Frohlich et al. | 2014 | Polarization of the effects of autoimmune and neurodegenerative risk alleles in leukocytes. Science | http://classic.science-mag.org/content/344/ 6183/519.long | Science, 2014. 344 (6183): p. 519-23 |
| Westra H-J, Peters MJ, Esko T, Yaghootkar H, Schurmann C, Kettunen J, Christiansen MW, Fairfax BP, Schramm K, Powell JE, Zhernakova A, Zhernakova DV, Veldink JH, Van den Berg LH, Karjalainen J, Withoff S, Uitterlinden AG, Hofman A, Rivadeneira F, 't Hoen PAC, Reinmaa E, Fischer K, Nelis M, Milani L, Melzer D, Ferrucci L, Singleton AB, Hernandez DG, Nalls MA, Homuth G, Nauck M, Radke D, Völker U, Perola M, Salomaa V, Brody J, Suchy-Dicey A, Gharib SA, Enquobahrie DA, Lumley T, Montgomery GW, Makino S, Prokisch H, Herder C, Roden M, Grallert H, Meitinger T, Strauch K, Li Y, Jansen RC, Visscher PM, Knight JC, Psaty BM, Ripatti S, Teumer A, Frayling TM, Metspalu A, van Meurs JBJ, Franke L | 2013 | Systematic identification of trans eQTLs as putative drivers of known disease associations | http://www.nature.com/ ng/journal/v45/n10/full/ ng.2756.html | Nat Genet, 2013. 45 (10): p. 1238-43 |
| The 1000 Genomes Project Consortium | 2015 | A global reference for human genetic variation | http://www.nature.com/ nature/journal/v526/ n7571/full/nature15393. html | Nature, 2015. 526 (7571): p. 68-74 |
| Lappalainen et al. | 2013 | Transcriptome and genome sequencing uncovers functional variation in humans | http://www.geuvadis. org/web/geuvadis/RNA-seq-project | Nature, 2013. 501 (7468): p. 506-511 |

| Pazin MJ | 2015 | Using the ENCODE Resource for Functional Annotation of Genetic Variants. | https://www.encodepro-ject.org/ | Cold Spring Harb Protoc, 2015. 2015(6): p. 522-36 |
| Boyle AP1, Hong EL, Hariharan M, Cheng Y, Schaub MA, Kasowski M, Karczewski KJ, Park J, Hitz BC, Weng S, Cherry JM, Snyder M | 2012 | Annotation of functional variation in personal genomes using RegulomeDB | http://regulomedb.org/ | Genome Res, 2012. 22(9): p. 1790-7. |

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
