## [Decision Letter]

Thank you for submitting your work entitled "Regulatory polymorphisms in XL9 modulate the expression of HLA class II molecules and promote autoimmunity" for consideration by *eLife*. Your article has been reviewed by two peer reviewers, and the evaluation has been overseen by a Reviewing Editor and Mark McCarthy as the Senior Editor.

The reviewers have discussed the reviews with one another and the Reviewing Editor has drafted this decision to help you prepare a revised submission. As you can see the reviews and the evaluation is positive, but there are a number of points that need to be addressed before we can accept the manuscript. In line with *eLife* principles, rather than send individual reviews back, the editorial team has compiled the issues raised by the reviewers into the summary list below.

Summary:

Through targeted deep sequencing of 28 risk loci for SLE the authors demonstrate that haplotypes often contain multiple putative functional variants, such as those altering transcription factor binding sites. Importantly, haplotypes account for a larger proportion of disease risk than the originally defined GWAS hits. The paper contains a number of important findings.

The authors report association between SLE disease risk haplotypes in the HLA Class II region with both expression QTLs and large differences in surface protein expression levels on antigen presenting cells. This has implications for our understanding of how risk alleles drive autoimmunity. Analysis of transcription at risk haplotype-containing genes with the XL9 region indicates that functional variants modify binding of IRF4. Expression of HLA class II alleles (D alleles) is higher in risk haplotypes than protective ones. This is a novel insight into the nature of genetic associations within the HLA. While the bulk of the description of the data is reserved for the STAT4 locus and the HLA, and appropriately so, there are also detailed analyses provided for the other loci studied, which is of significant value. Altogether this study provides important new information on the genetics of SLE.

Essential revisions:

1) For the RNA-seq analysis throughout, normalized RPKMs (normalized for the length of the gene) should be used, not raw RPKMs. More information on the methods used (average depth, read length, single or paired end, mapping of reads, software used to compute RPKM values) are needed. Report how many samples were excluded for <25X coverage, how many excluded for HWD in controls. How many samples were at start and how many at the end after filtering? (For example: in Figure 4, what is being plotted: mean or median RPKM? What are sample sizes for each genotype? Please show p-values for all).

2) In several instances the authors claim either confirmation or stronger association for some of the known loci studied. To what extent are the samples independent from those used in previous GWAS and association studies?

3) Could the authors comment on previous studies showing the effects of compound heterozygotes of HLA DR2 and DR3 haplotypes in the genetic association of SLE and the relationship with XL9? One line can be added to mention compound heterozygozity.

4) Figure 6 – Some attention is needed on this figure. Mismapping of RNA-seq reads is a known problem for HLA eQTL analysis and may lead to false positives. Which steps have you taken to avoid this? Panel E: the central figure in (iv), the legend misses the second allele (A). Also keep strands the same in this figure (ii vs. iii). Panel A: What cell type for the DNAse hypersensitivity track? The SNPs shown in 6E are not the same SNPs that reside in the IRF4 binding motif (6B), while the figure with the arrow from panel C to E seems to imply a direct connection. Please clarify.

5) Figure 7: Provide a key for the heat map, and p-values for differences. Was quantitative RNA expression performed on these same donors? If so, please include. Please include in legend or on the figure the actual MFIs for the various genotypes. Please also clarify that R848 is a TLR7/8 ligand.

---

## [Author Response]

*Essential revisions:*

*1) For the RNA-seq analysis throughout, normalized RPKMs (normalized for the length of the gene) should be used, not raw RPKMs. More information on the methods used (average depth, read length, single or paired end, mapping of reads, software used to compute RPKM values) are needed. Report how many samples were excluded for <25X coverage, how many excluded for HWD in controls. How many samples were at start and how many at the end after filtering? (For example: in Figure 4, what is being plotted: mean or median RPKM? What are sample sizes for each genotype? Please show p-values for all).*

We apologize for the confusion, but the RPKM values presented in the manuscript were normalized for the length of the gene. We have used the term “normalized RPKM” in the revised figures and the manuscript text as suggested by the reviewers.

We have added details and additional information throughout the Methods section. We expanded the description of the protocol used for the culturing and production of monocyte-derived dendritic and macrophages, together with an expanded description of the methods for RNA-SEQ data production and analysis, including read depth, read length, read mapping and software used for these RNA-Seq analyses to the Methods section.

For the filtering of samples, the study started with 1775 samples, all of which were sequenced following targeted enrichment. Of these, 88 samples had missing case/control status information and 249 were PCA outliers for the HapMap CEU reference sample in principal component analysis, most of which were self designated as African American or Hispanic. Of the remaining 1438 PCA pass samples, 11 and 5 were excluded due to poor call rate (<85%) and being duplicate samples, respectively. In addition, 73 samples were excluded due to poor sequencing fold coverage (<25x, n=54) or significant p value (p>0.001) of HWE in controls (n=19). Thus, final set of 1349 samples which included 773 SLEs and 576 normal controls passed all quality criteria and were used for association analysis. This information is incorporated into the Methods section in the subsection “Defining high quality variants in the EA population”.

Mean RPKM values are plotted for all graphs of eQTL data, including in Figure 4 and the sample size of each genotype is given in parentheses for each genotype along the X axis.P-values are now shown for both the plots in the revised manuscript.

*2) In several instances the authors claim either confirmation or stronger association for some of the known loci studied. To what extent are the samples independent from those used in previous GWAS and association studies?*

More than 50% of SLE cases and all of the control samples used in the present study were new recruitments and have not been used in any previous association or GWAS on SLE. The remainders were provided by Dr. Betty Tsao and were used in our original GWAS study by Harley et al. (2008). This information has been added to the subsection “Targeted sequencing of SLE risk loci” in the Methods section.

*3) Could the authors comment on previous studies showing the effects of compound heterozygotes of HLA DR2 and DR3 haplotypes in the genetic association of SLE and the relationship with XL9? One line can be added to mention compound heterozygozity.*

HLA is a well-established genetic risk locus associated with SLE. Many MHC gene studies and GWAS have shown strong association of HLA-DR2 and HLA-DR3 alleles with SLE. A study by Graham et al. (2007) have explored the effects of different combinations of HLA-class II alleles in the context of SLE and its component phenotypes and showed that DR3 allele pose stronger SLE risk than DR2 allele. The study also found that individuals homozygous for DR3 or compound heterozygote for DR3 and DR2 allele demonstrated highest disease risk. As shown in Figure 17 below, this is also true for comparisons of the regulatory haplotypes for XL9 that are in strongest LD with DR3 (HAP3) and DR2 (HAP2). We have added sentences concerning these issues and added the appropriate citation in the subsection “HLA-D polymorphisms, antigen presentation pathways, and autoimmune disease” in the Discussion section.

Author response image 1.A comparison of OR for individuals homozygous for XL9 HAP3 (DR3), XL9 HAP2 (DR2), and HAP2/HAP3 heterozygotes.**DOI:**
http://dx.doi.org/10.7554/eLife.12089.024

*4)*
Figure 6 – *Some attention is needed on this figure. Mismapping of RNA-seq reads is a known problem for HLA eQTL analysis and may lead to false positives. Which steps have you taken to avoid this? Panel E: the central figure in (iv), the legend misses the second allele (A). Also keep strands the same in this figure (ii vs. iii). Panel A: What cell type for the DNAse hypersensitivity track? The SNPs shown in 6E are not the same SNPs that reside in the IRF4 binding motif (6B), while the figure with the arrow from panel C to E seems to imply a direct connection. Please clarify.*

We agree with the reviewers that mapping sequencing reads into the highly polymorphic HLA region is troublesome. We applied the following measures to control for false positive eQTLs for all of the MDM eQTLs reported in this manuscript (including HLA-DR and DQ in Figure 6): 1) High read depth of sequencing (>31,000,000 reads on average/sample) and removal of poor quality reads (still leaving >96% genome alignment on average of all pass filters reads for all samples in cohort); 2) Concordance rate for genomic versus RNA-Seq variant calls of >98% for heterozygous calls as a measure of mapping accuracy; and 3) Validation of previously published eQTLs in our monocytes derived macrophage data reported here.

To ensure the accurate alignments and mapping of RNA-Seq reads to the reference genome for the allelic bias studies of HLA-DR and DQ shown in Figure 6, we used CLC-Biosystems tools to map the RNA-Seq sequencing reads to the reference genome and validated all variant calls using additional variant calling software. As shown with representative data in Figure 6 (iii) and C(iv) in the manuscript and in the data shown below in Figure 18, the read depth for HLA was very high. Further, all variants called in the RNA-Seq datasets were coincided with and were confirmed by the variants called in the individual’s genomic sequence.

The most significant alignment issues involve reads mapping into the highly polymorphic exon 2 segments of HLA class II genes (which encodes the highly diversified peptide binding groove of HLA class II molecules). Therefore, for our analysis of allelic bias, we looked at multiple SNPs throughout exon 2 and throughout the much less polymorphic exon 3 (which encodes the membrane-proximal structural domains) of DRB1, DQB1, and DQA1. As shown in Figure 18 below, multiple variants in exon 2 and exon 3 were found to be in significant bias with all of these variants trending towards the same allelic imbalance. In this regard, the exon 3 variants, two of which were used for Figure 6Ciii-iv, showed more bias then many of the exon 2 variants. These results confirm the observed allelic bias and indicate that it is not only observed in the highly polymorphic exon 2, but is also found in variants in exon 3.

Author response image 2.Representative data from heterozygote samples used for analysis of allelic expression bias in Figure 6 iii and iv.Note that multiple variants in exon 2 and exon 3 showed allelic bias in the same direction, although the magnitude of the effect varied between SNPs, presumably due to variations in read depth at specific locations in the exons.In this regard, variations in the less polymorphic exon 3 showed a stronger allelic bias than those in exon 2, although all of these SNPs showed significant allelic bias in frequency within the RNA-seq data favouring the variants associated with the risk allele..**DOI:**
http://dx.doi.org/10.7554/eLife.12089.025

For the other points raised:

1) Figure 6 legend in Figure 6(iv) and strand order in the tables in Figure 6(ii) has been corrected as suggested.

2) The DNase hypersensitivity track shown in Figure 6 is generated based on 125 different cell types studied in the ENCODE project and is available on the UCSC human genome browser as an option. These include immune cells such as CD4+T cells, CD20+ B cells, lymphoblastic cells, monocytes, etc., which are important in the context of SLE disease. More details on cell types studied can be found at: https://www.encodeproject.org/search/?type=Biosample&organism.scientific_name=Homo+sapiens.

3) The IRF4 binding motif SNP is not shown here. But the RNAseq data shown in Figure 6(iv) is based on four independent human donors who were heterozygous for IRF4 SNPs. For simplicity, we used the peak tagging SNP to show the LD relationships between the specific coding variants and the risk and protective regulatory haplotypes. The arrow from Figure 6 to Figure 6 was intended to point to the observed biased in RNAseq data as modeled in Figure 6. To avoid the confusion, we have removed the arrow in the revised manuscript.

We have modified Figure 6 in the revised manuscript as discussed above and have expanded the text describing these experiments.

5) Figure 7: Provide a key for the heat map, and p-values for differences. Was quantitative RNA expression performed on these same donors? If so, please include. Please include in legend or on the figure the actual MFIs for the various genotypes. Please also clarify that R848 is a TLR7/8 ligand.

We have added a key for the heatmap and p-values for the differences in Figure 7 for the revised manuscript. Also, RNAseq expression data from the same donors and expression of HLA class II genes in both the donors are now included in revised Figure 7. In addition, we have included graphs of the MFIs for all of the donors and an independent replicate of the HLA-DR donors. Reference to R848 as TLR7/8 ligand is given and stated in the figure legend. The text has been modified to add these additional details.